# Adipocyte deletion of the oxygen-sensor PHD2 sustains elevated energy expenditure at thermoneutrality

Rongling Wang [1], Mario Gomez Salazar[1], Iris Pruñonosa Cervera[1], Amanda Coutts [2], Karen French[1], Marlene Magalhaes Pinto[1], Sabrina Gohlke[3], Ruben García-Martín [4], Matthias Blüher [5], Christopher J. Schofield [6], Ioannis Kourtzelis [7], Roland H. Stimson [1], Cécile Bénézech[1], Mark Christian [2], Tim J. Schulz [3,8], Elias F. Gudmundsson [9], Lori L. Jennings [10], Vilmundur G. Gudnason [9,11], Triantafyllos Chavakis [1,8,12,13], Nicholas M. Morton[1,2], Valur Emilsson [9,11] & Zoi Michailidou [1,2] ✉

Enhancing thermogenic brown adipose tissue (BAT) function is a promising therapeutic strategy for metabolic disease. However, predominantly thermoneutral modern human living conditions deactivate BAT. We demonstrate that selective adipocyte deficiency of the oxygen-sensor HIF-prolyl hydroxylase (PHD2) gene overcomes BAT dormancy at thermoneutrality. Adipocyte-PHD2-deficient mice maintain higher energy expenditure having greater BAT thermogenic capacity. In human and murine adipocytes, a PHD inhibitor increases *Ucp1* levels. In murine brown adipocytes, antagonising the major PHD2 target, hypoxia-inducible factor-(HIF)−2a abolishes *Ucp1* that cannot be rescued by PHD inhibition. Mechanistically, PHD2 deficiency leads to HIF2 stabilisation and binding of HIF2 to the *Ucp1* promoter, thus enhancing its expression in brown adipocytes. Serum proteomics analysis of 5457 participants in the deeply phenotyped Age, Gene and Environment Study reveal that serum PHD2 associates with increased risk of metabolic disease. Here we show that adipose-PHD2-inhibition is a therapeutic strategy for metabolic disease and identify serum PHD2 as a disease biomarker.

Several elegant studies have identified approaches to activate thermogenic brown adipose tissue (BAT) and enhance metabolism[1–3]. However, the majority of these studies were conducted on rodents at ambient temperature (19–21 °C), a condition where BAT is already active[4–6]. Consequently, the human translation of these studies is questionable, because humans predominantly live in thermoneutral conditions by using clothing or other means to maintain thermal homeostasis[7]. One highly upregulated gene in adipose tissue during cold exposure of rodents is the hypoxia inducible transcription factor (HIF)-2a[8], suggesting that the hypoxia signalling pathway is involved in BAT activation. The oxygen-sensing HIF-prolyl hydroxylases (PHDs)

are central regulators of the hypoxic response[9,10]. PHD2 is reported to be the most important oxygen sensor and is a negative regulator of HIF-a activity in normoxia[9,10]. PHD2 hydroxylates key proline residues on HIF-a isoforms in normoxia, leading to their proteasomal degradation[9,10].

PHD inhibitors are of increasing medical significance and have been approved for the treatment of renal anaemia in chronic kidney disease (CKD)[11] in China and Japan[12,13] and very recently in Europe by the European Medicine Agency[14]. Previously, we and others, have shown that models of whole-body or adipose selective deletion of the dominant, ubiquitously expressed PHD2 isoform, induced protection

from metabolic disease, in part by reducing plasma lipid levels[15,16]. Here we hypothesized that the lipid-lowering effects of PHD2 deficiency is due to enhanced BAT thermogenesis. We tested this hypothesis, using an adipose tissue (white and brown) selective genetic mouse model of PHD2 deletion (P2KO[ad]) under the more human translationally aligned thermoneutral housing conditions. To address the relevance of PHD2 in human metabolic disease traits, we interrogated the unique large-scale serum proteomics and deep metabolic and anthropometric phenotyping data from the Age/Gene Environment Susceptibility study (Icelandic Heart Association)[17,18].

## Results

### Loss of adipocyte *Phd2* sustains energy expenditure at thermoneutrality in both male and female mice

Housing of mice at thermoneutrality (TN; ~28–30 °C) results in a marked decrease in energy expenditure and loss of BAT function[19,20]. This aligns rodent metabolism more closely with that of humans, providing a better preclinical modelling system[5–7,19–21]. We tested whether the metabolic advantage observed with adipose-specific PHD2 deletion previously[16,22], was retained at TN (28–29 °C), thus indicating an effect independent from, the tonic, metabolically protective activation of BAT function found at room temperature (RT;19–21 °C) in mice.

Adiponectin–Cre[23] mice were crossed with PHD2[fl/fl] mice[16] to delete PHD2 (referred as P2KO[ad]) in white (mean *Phd2* mRNA levels ±SEM: Control:1.8 ± 0.23 arb. units ($n = 5$) vs P2KO[ad]: 0.14 ± 0.14 arb. units ($n = 6$), $P < 0.0001$) and brown adipocytes (Supplemental Fig. 1A). This approach targets all white (WAT), beige and BAT adipocytes[24] and thus reflects a generalised adipose tissue deletion not restricted only to the rodent-specific BAT depot. P2KO[ad] mice had >85% reduction in *Phd2* mRNA levels compared to control littermates, and consequently elevated levels of the validated HIF-target genes, *Phd3* (fourfold, $P = 0.026$) and *Vegfa* (1.6-fold, $P = 0.018$) in BAT (Supplemental Fig. 1A). *Vegfa* mRNA levels in inguinal white adipose tissue (iWAT, mean ± SEM: Control: 0.38 ± 0.07 arb. units ($n = 5$) vs P2KO[ad]: 0.73 ± 0.12 arb. units ($n = 6$), $P = 0.045$) of P2KO[ad] mice were higher compared to control littermates. In P2KO[ad] BAT, HIF-1a and HIF2a levels were stabilised, as expected (Supplemental Fig. 1B), irrespective of oxygen levels in the tissue. These results are consistent with enhanced vascularization in the adipose tissues in this model as we previously showed in WAT irrespective of ambient temperature[16]. P2KO[ad] mice initially housed at room temperature (RT; 20–21 °C) were then examined at acute thermoneutrality (TN; 28–29 °C, 7 days). Unexpectedly, despite similar body weights (Fig. 1A), P2KO[ad] mice had relatively higher energy expenditure (EE), especially in the dark phase (Fig. 1B) than control littermates. Overall, the P2KO[ad] mice showed a skew towards lipid utilization (lower RER) and this was statistically significant at room temperature (during the whole day) and in the dark phase at thermoneutrality (Fig. 1C). There were no differences in physical activity levels (Fig. 1D). Presumably to compensate for their higher EE at TN, P2KO[ad] mice exhibited higher food intake (Fig. 1E) compared to their littermate controls. Switching male mice from RT to TN, led to increased fat mass deposition in both genotypes (Fig. 1F), such that at thermoneutrality, body gross composition, determined by TD-NMR, was similar between the genotypes (lean and fat mass, Fig. 1F). However, the control littermate mice lost BAT mass at TN, whereas the P2KO[ad] mice, maintained BAT mass (Fig. 1G) and had significantly greater BAT mass (Fig. 1G). White adipose depots (Fig. 1H) and liver mass (mean ± SEM: Control:1.33 ± 0.17g vs P2KO[ad]: 0.42 ± 0.20 g ($n = 7$), $P = 0.36$) were comparable between genotypes.

Energy homeostasis differs between sexes. Rodent and human studies suggest that females have higher capacity to increase mass and thermogenic activity of adipose tissue[25,26]. To test whether adipose-*Phd2* deletion would be advantageous in both sexes as a target to increase energy expenditure, we set paired experiments, switching

from RT to TN housing, in female P2KO[ad] (*Egln1*, 90% knock down achieved in female adipose tissue, Supplemental Fig. 1C) and control littermates. As in males, P2KO[ad] female mice had similar body weights at RT and TN (Fig. 2A) to control littermates but retained higher energy expenditure at thermoneutrality (Fig. 2B), but no significant differences in substrate utilization (Fig. 2C) or physical activity (Fig. 2D). Switching to thermoneutrality led to reduced food intake in the control female mice (Fig. 2E). Female P2KO[ad] mice retained similar food intake in both temperatures (Fig. 2E), in contrast to P2KO[ad] males that increased food intake at TN. Body composition in females (fat and lean mass) measured by TD-NMR was unchanged in thermoneutrality (Fig. 2F), in contrast to males (Fig. 1F). However, in females, BAT mass was specifically increased in both genotypes (Fig. 2G) with no changes observed in WAT mass (Fig. 2H) or other metabolic tissues like the liver (mean ± SEM: Control:1.2 ± 0.09 g vs P2KO[ad]: 1.3 ± 0.04 g ($n = 7–9$), $P = 0.37$). Taken together these data show that both male and female mice lacking adipose-*Phd2* sustain higher energy expenditure at thermoneutrality.

### Loss of adipose-*Phd2* facilitates better metabolic health at thermoneutrality

Poor metabolic health is associated with low BAT activity[27]. Metabolically, the main effect of adipose-*Phd2* deletion was on plasma lipids in both sexes. Specifically, P2KO[ad] mice had significantly lower circulating glycerol and non-esterified fatty acids (NEFA) levels compared to control mice (Fig. 3A–D) at RT. Male P2KO[ad] mice retained lower levels of plasma lipids when housed at TN (Fig. 3A, C). Glucose levels were not affected by genotype or housing temperature (Fig. 3E, F). Insulin levels were similar in control and P2KO[ad] male mice (Fig. 3G). However, female P2KO[ad] mice had lower plasma insulin levels compared to control littermates in both RT and TN (Fig. 3H), despite similar glucose levels (Fig. 3F).

### Loss of adipocyte *Phd2* mediates retention of BAT function at thermoneutrality

Although energy expenditure was increased in both male and female mice, and irrespective of genotype, females retained BAT mass at thermoneutrality, only male P2KO[ad] mice had significantly greater BAT mass than their control littermates. At a molecular level, as expected, switching mice from RT to TN, decreased mRNA levels of the thermogenic gene *Ucp1* in BAT (Supplemental Fig. 1D). However, deletion of adipocyte *Phd2* led to sustained (and higher compared to control mice) *Ucp1* mRNA levels and the BAT-activating adrenergic receptor, *Adrb3*, in BAT (Supplemental Fig. 1E) and in iWAT (Supplemental Fig. 1F, G) in male mice at thermoneutrality. There were sex-differences on this response, as female mice retained only higher BAT *Adrb3* levels (Supplemental Fig. 1E) in both housing conditions *but* only sustained significantly higher *Ucp1* mRNA levels in iWAT (Supplemental Fig. 1F) during TN. As the effects were more pronounced and consistent in the BAT and iWAT of male mice, we continued further investigations in male mice.

The main characteristic during BAT activation in cold exposure or β-adrenergic stimulation is BAT hyperplasia and/or hypertrophy[28,29]. To investigate the cellular basis for the unexpected resistance of BAT to regress at TN in male P2KO[ad] mice, we performed histological analysis. P2KO[ad] mice had bigger adipocytes, in the range of 450–1000 μm², but similar brown adipocyte numbers to control mice (Fig. 4A, B), suggesting hypertrophy in BAT. To provide additional insights into BAT function, we quantified UCP1+ cells at thermoneutrality using antibody staining as a proxy for functional BAT. P2KO[ad] mice exhibited more UCP1 expressing cells (Fig. 4C, D and Supplemental Fig. 2A) in BAT than control littermates and more ki67+ cells (Fig. 4C, D and Supplemental Fig. 2B), suggesting higher proliferation rates. Higher protein UCP1 levels in BAT in P2KO[ad] mice were confirmed by western blot analysis (Supplemental Fig. 2D).

Previously, we have shown that deletion of *Phd2* leads to better WAT vascularization (increased *Vegfa* mRNA/CD31

immunoreactivity)[16] at ambient housing conditions (21 °C). We confirmed here that *Vegfa* mRNA levels remain higher at thermoneutrality in *P2*KO[ad] mice BAT (Supplemental Fig. 1A). Immunofluorescence for isolectin IB4, a vessel marker, also confirmed more extensive BAT vascularization in *P2*KO[ad] mice (Fig. 4C, D and Supplemental Fig. 2C). Taken together this data suggest, the loss of adipose-*Phd2* permits better BAT remodelling at thermoneutrality.

## Loss of adipocyte-*Phd2* leads to higher maximal response to β3-adrenergic stimulation

To further examine BAT/beige function we performed a test of pharmacological BAT activation using a β3-adrenergic receptor agonist (CL 316,243; CL) in a paired experiment. *P2*KO[ad] mice had a higher response to adrenergic stimulation when examined at 28 °C (Fig. 5). Despite similar body weights (C vs KO: 27.5 ± 0.4 g vs 27.9 ± 0.5 g,

$P = 0.47$), *P2*KO[ad] mice had significantly higher energy expenditure immediately post CL administration that was sustained throughout the rest of the day (Fig. 5A), and this was not due to higher activity levels (Fig. 5B). CL induced a switch to utilising lipids in both genotypes with a significantly lower RER in *P2*KO[ad] mice (Fig. 5C). Additionally, CL-induced significantly higher plasma NEFA release in the *P2*KO[ad] mice (Fig. 5D) and higher *Adrb3* and *Ucp1* mRNA expression in both BAT and iWAT (Fig. 5E). Importantly, this was confirmed at the protein level, as UCP1 was higher in the *P2*KO[ad] BAT after CL stimulation (Fig. 5F).

## Loss of adipose-*Phd2* retains higher EE after diet induced obesity at thermoneutrality

High fat diet (HFD) feeding and thermoneutrality independently reduce energy expenditure and the thermogenic gene expression program in BAT[20]. Feeding HFD at thermoneutrality, as a more

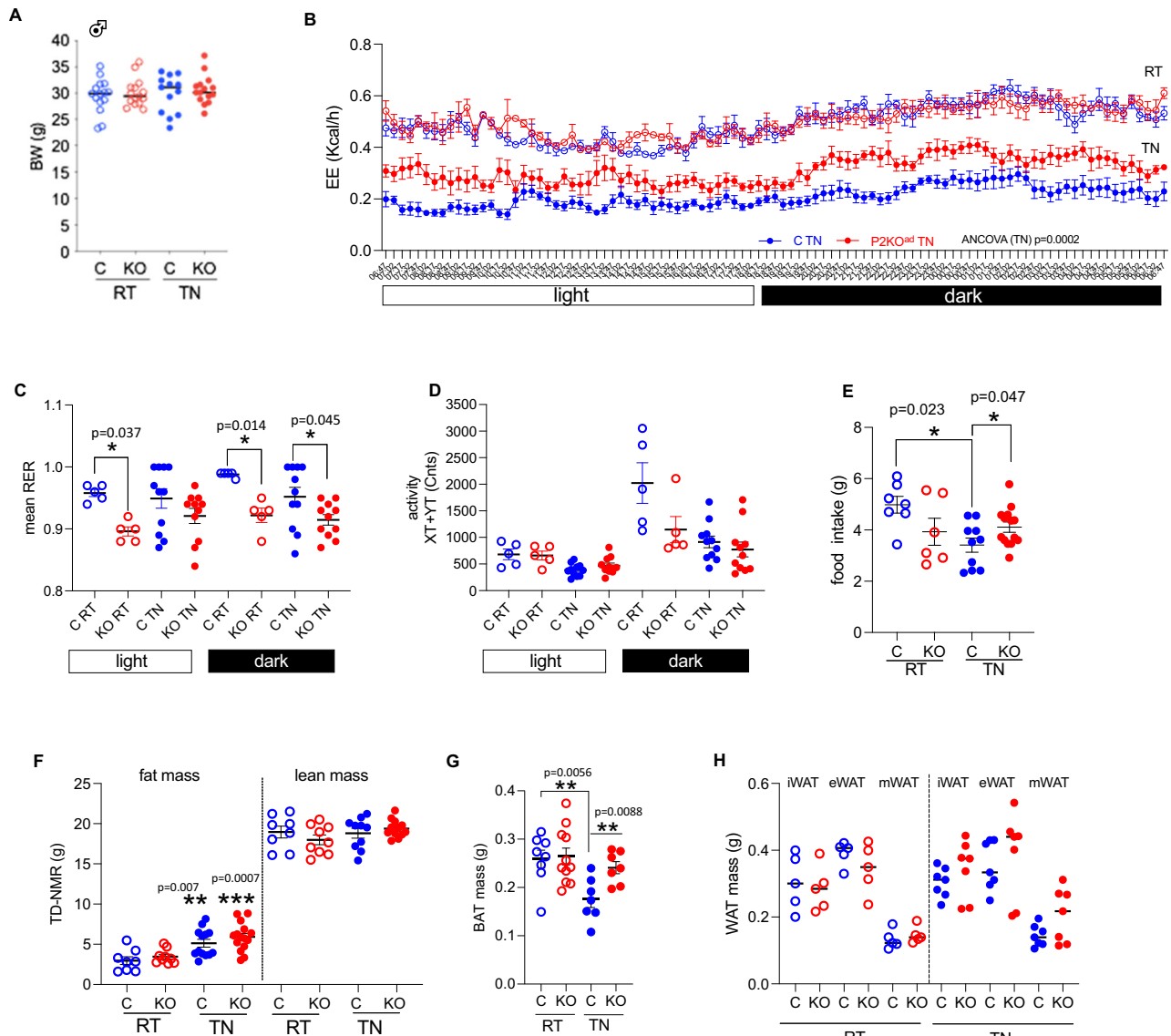

**Fig. 1 | Adipose-*Phd2* deletion in male mice sustains higher energy expenditure at thermoneutrality. A** *P2*KO[ad] (KO, red) mice housed at RT (21 °C, *n* = 15) or acutely switched to TN (28–29 °C, 7 days) sustain similar body weight (BW) to control (C, blue, *n* = 13) littermates. **B** *P2*KO[ad] mice have higher energy expenditure (EE) at TN (*n* = 6/group) and **C** lower respiratory exchange ratio (RER, RT: *n* = 5/group; TN: *n* = 11/group). **D** Similar activity levels in both genotypes (RT: *n* = 5/group; TN: *n* = 11/group). **E** KO (RT: *n* = 6; TN: *n* = 14) eat more than C (RT: *n* = 7; TN:

*n* = 14) mice at TN. **F** Despite similar fat and lean mass in both genotypes measured by time-domain (TD) NMR (RT: n = 8/group; TN control: *n* = 10; TN KO: *n* = 11), **G** *P2*KO[ad] mice have bigger BAT mass (C RT: *n* = 8; KO RT: *n* = 11; C TN: *n* = 7; KO TN: *n* = 7) but **H** similar WAT mass (RT: *n* = 5/group; TN: *n* = 7/group), biological replicates. Data are presented as mean +/− SEM. *P < 0.05, **P < 0.01 by Student *t* test (two-tailed). For EE, ANCOVA was performed with BW as covariance. Source data are provided as a Source Data file.

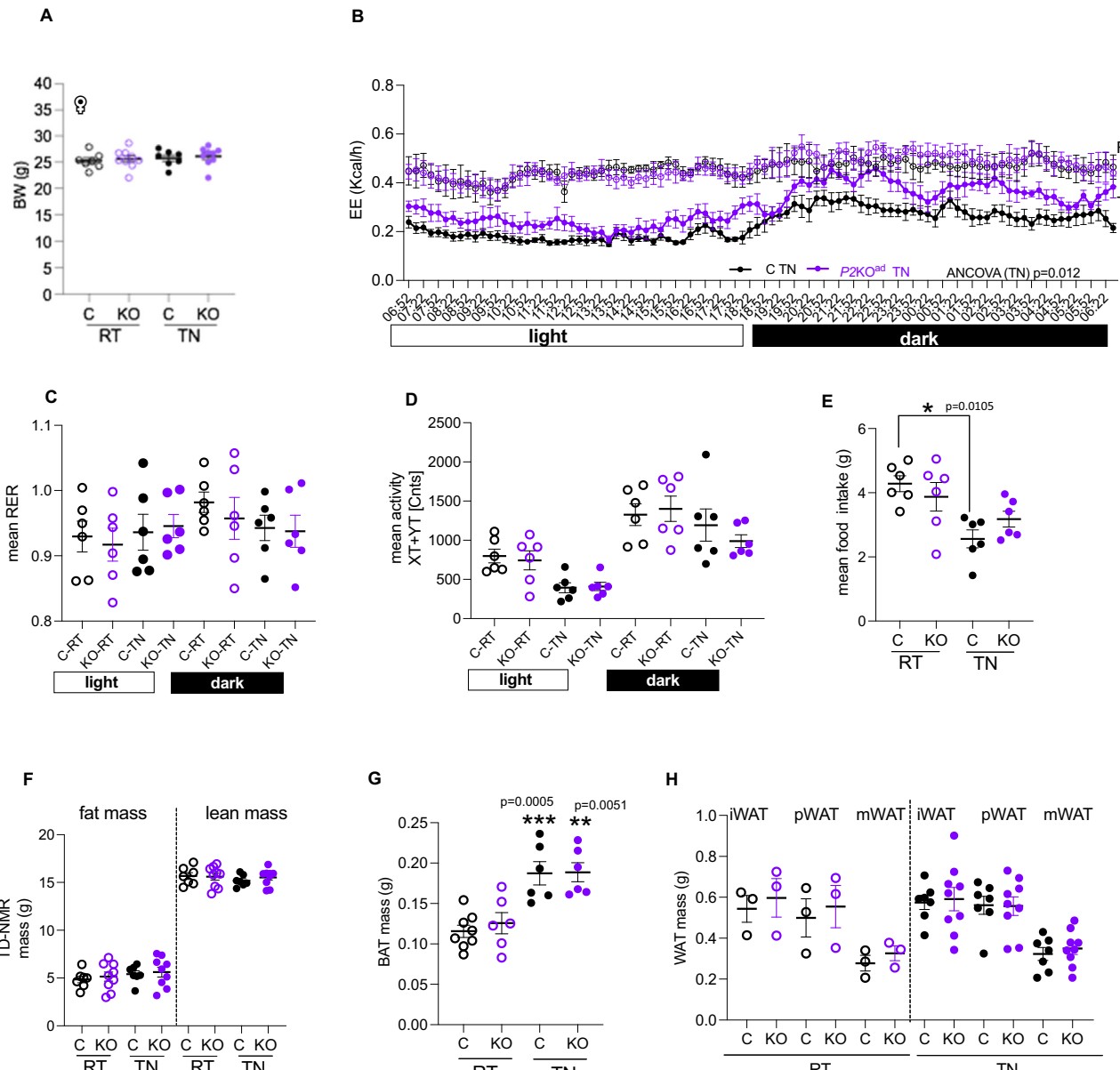

**Fig. 2 | Adipose-*Phd2* deletion in females sustains higher energy expenditure at thermoneutrality. A** *P2*KO[ad] (KO, purple) mice housed at 20–21 °C (RT, *n* = 9) when acutely switched to 28–29 °C (TN, 7 days) sustain similar body weight (BW) to control (C, black, *n* = 7) littermates but **B** have higher energy expenditure (*n* = 6/group). **C** Similar respiratory exchange ratio (*n* = 6/group) and **D** activity levels (*n* = 6/group). **E** Control mice reduce, but KO sustain food intake at TN (*n* = 6/group). **F** Similar fat and lean mass measured by TD-NMR (C: *n* = 7 and KO: *n* = 9) in both genotypes. **G** Similar BAT mass (*** indicates comparisons of RT with TN; RT C: *n* = 8, KO: *n* = 6 and TN: *n* = 6/group) and **H** WAT mass (i inguinal, p parametrial, m mesenteric, RT: *n* = 3/group; TN C: *n* = 7 and KO: *n* = 9) in both genotypes. *n* indicates biological replicates. Data are presented as mean +/− SEM. *P < 0.05, **P < 0.01, ***P < 0.001 by paired (**A**–**F**) or unpaired (**G**, **H**) Student *t* test (two-tailed). For EE, ANCOVA was performed with BW as covariance. Source data are provided as a Source Data file.

translationally aligned model for diet-induced obesity (DIO), further suppresses energy expenditure and BAT function[20]. To test whether the higher energy expenditure at TN seen in *P2*KO[ad] can be sustained during HFD, we fed male mice HFD for 8-weeks while housed at TN. Despite similar body weights (Fig. 6A) and body weight gain (Supplemental Fig. 3A) after 8 weeks of HFD in both genotypes, *P2*KO[ad] mice retained higher energy expenditure during HFD compared to their control littermates (Fig. 6B). No significant differences were detected in substrate utilization (Fig. 6C) or physical activity (Fig. 6D) between control and *P2*KO[ad] mice. Unfortunately, we could not accurately record food intake in this experiment as most mice shredded their HFD. Analysis of the food intake data from the first couple of days of HFD did not show any significant differences (Supplemental Fig. 3B).

Fat mass gain after HFD (Fig. 6E), including fresh WAT (Fig. 6F) and BAT (Fig. 6G) weight, change in lean mass (Supplemental Fig. 3C) and liver weights (Supplemental Fig. 3D) were similar in both genotypes. Although their glucose levels were similar (Fig. 6H), *P2*KO[ad] mice retained lower plasma NEFA (Fig. 6I). Fasting plasma insulin levels were similar, but in the fed state, the *P2*KO[ad] mice did not significantly increase insulin levels in contrast to control littermates (Fig. 6J).

One of the hallmarks of DIO is monocyte recruitment in adipose tissue and an increase of inflammatory macrophages[30], therefore we performed analysis of immune cells in blood and WAT (Supplemental Fig. 4A). Blood and WAT neutrophil numbers were similar in control and *P2*KO[ad] mice (Supplemental Fig. 4B, D). However, *P2*KO[ad] mice had reduced numbers of blood monocytes (Supplemental Fig. 4C), and a

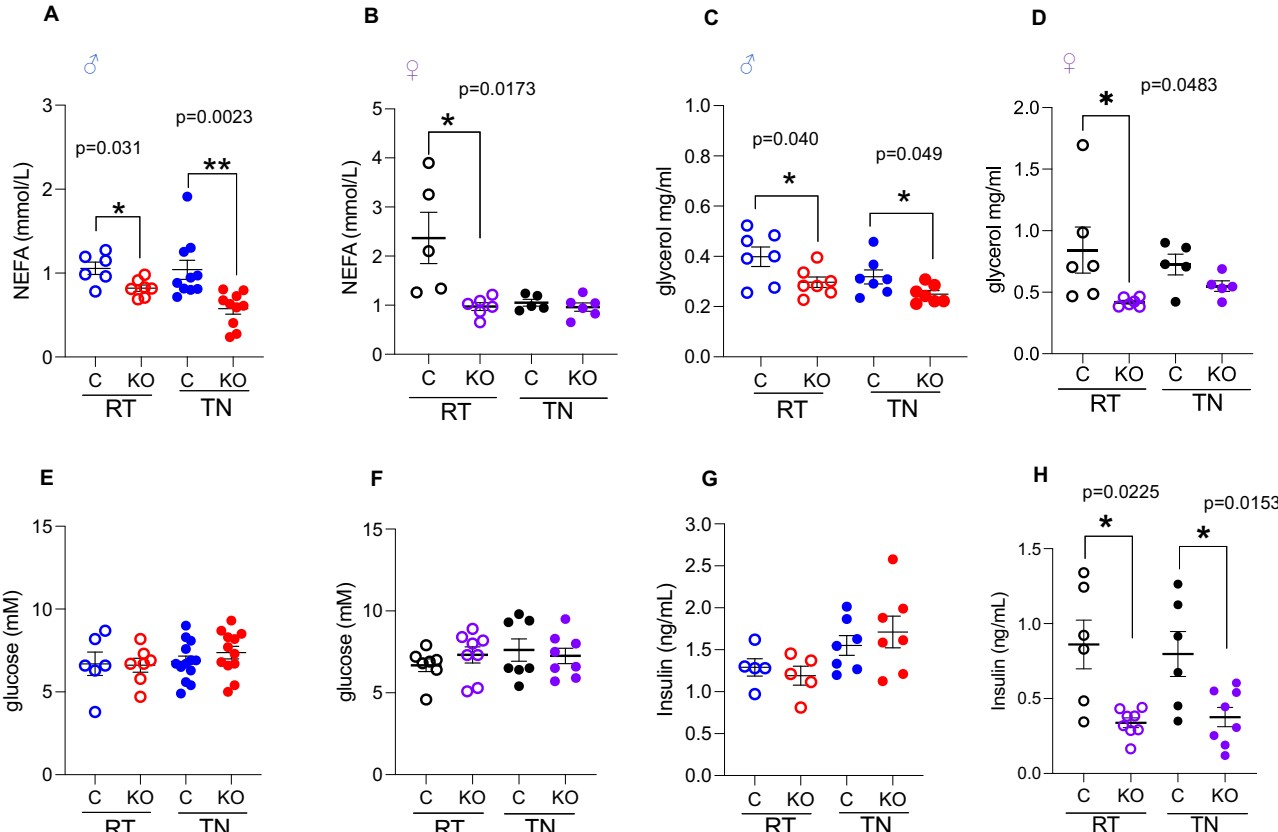

**Fig. 3 | Metabolic parameters in male and female *P2*KO^ad mice housed at room temperature (RT) and thermoneutrality (TN).** **A**, **B** Plasma non-esterified fatty acids, **C**, **D** glycerol, **E**, **F** glucose and **G**, **H** insulin levels. Males, NEFA: control RT $n = 6$, TN $n = 10$, KO RT $n = 7$ TN $n = 10$, glycerol: $n = 7$/group, glucose: control RT $n = 6$ TN $n = 13$, KO RT $n = 7$, TN $n = 12$, insulin: RT $n = 5$/group, TN $n = 7$/group).

Females, NEFA: control $n = 5$, KO $n = 6$; glycerol: control $n = 5$, KO $n = 5$; glucose: control $n = 7$, KO $n = 8$; insulin: control $n = 6$, KO $n = 8$). Data are presented as mean +/− SEM. *N* represents biological replicates. *$P < 0.05$, **$P < 0.01$ by Student $t$ test (two-tailed). Source data are provided as a Source Data file.

trend for lower WAT monocytes (Supplemental Fig. 4E). No significant differences were detected in resident (Supplemental Fig. 4F) and inflammatory macrophages (Supplemental Fig. 4G) in WAT. *P2*KO^ad mice had lower mRNA levels of the monocyte chemotactic protein *Ccl2* (Supplemental Fig. 4H).

It is well established that HFD increases HIF-signalling in adipose tissue[31–33], therefore it is not surprising that the HFD challenge reduced the differences detected with thermoneutrality alone. Indeed, the mRNA levels of the established HIF-signature genes (*Epas1* and *Vegfa*) after HFD are identical between genotypes (Fig. 6K). This could explain the less pronounced differences between genotypes after HFD at TN. Despite this, *P2*KO^ad mice still retained modestly higher *Adrb3* mRNA, but not *Ucp1* levels in the BAT (Fig. 6K).

### Pharmacological PHD inhibition induces UCP1 in murine and human adipocytes in vitro

Clinically, the most advanced developed inhibitors approved by EMA for renal anaemia treatment in chronic kidney disease, are targeting all 3 PHD isoforms (pan-PHD inhibitors; PHDi). PHDi leads to HIF-1a and HIF-2a stabilization in various experimental models and tissues[16,22,34–36] including adipose tissue[16,22,35]. Because the PHD-HIF pathway contains several components that makes dissection of major contributors on the observed effect challenging, we interrogated RNA-sequencing data on the expression levels of all the major components of the HIF/PHD pathway (*Phd1-2-3*, *Hif1-2*) in BAT in mice at room temperature (Supplemental Fig. 5A). This directly comparable data show that in the BAT of 10-week old mice housed at room temperature, *Phd2* expression is more than 5-fold higher than *Phd1*, or *Phd3*. Similarly, *Hif2* is 10-fold

higher than *Hif1* expression. In addition, exposure to acute cold, does not regulate brown adipose tissue *Hif1* mRNA levels, in contrast to *Hif2* mRNA *levels*, which are elevated (Supplemental Fig. 6B). Importantly, we found that in thermoneutrality, *P2*KO^ad *Hif2* levels in BAT are similar to that seen in cold-treated controls (Supplemental Fig. 5B). As PHD2 and HIF2 are the only 2 components of the HIF pathway highly enriched in BAT, we focused on their effects on UCP1 regulation.

To address clinical significance, we tested whether FG2216 (a small molecule pan PHDi) directly could induce UCP1 expression in vitro as we have seen in the genetic model of adipose-specific PHD2 deletion. In parallel, we determined whether the PHD2 isoform is mainly responsible for the observed effect on UCP1 expression. To do this we treated control and *P2*KO^ad brown adipocytes with FG2216 (Fig. 7A). Basally, *P2*KO^ad brown adipocytes had already higher *Ucp1* expression than untreated control. FG2216 increased *Ucp1* mRNA levels in control brown adipocytes to similar levels seen in *P2*KO^ad brown adipocytes (Fig. 7A). However, FG2216, did not further increase *Ucp1* in *P2*KO^ad brown adipocytes, suggesting that inhibition of *Phd1* and *Phd3* was not enhancing the effect.

To confirm the PHDi effect seen in primary control brown adipocytes, we further tested a mouse immortalized BAT cell line[37], that has comparable *Ucp1* mRNA levels to mice at room temperature conditions (Supplemental Fig. 6). Treatment with FG2216, increased both UCP1 protein (Fig. 7B) and mRNA levels (Fig. 7C). The HIF-2a, and not the HIF-1a, signalling pathway was implicated as important for BAT function during cold-exposure ([38,39] and Supplemental Fig. 5B). To confirm we tested whether the PHD-HIF-2a axis underpinned the effects of the PHDi (FG2216). The effect of PHDi on *Ucp1* and

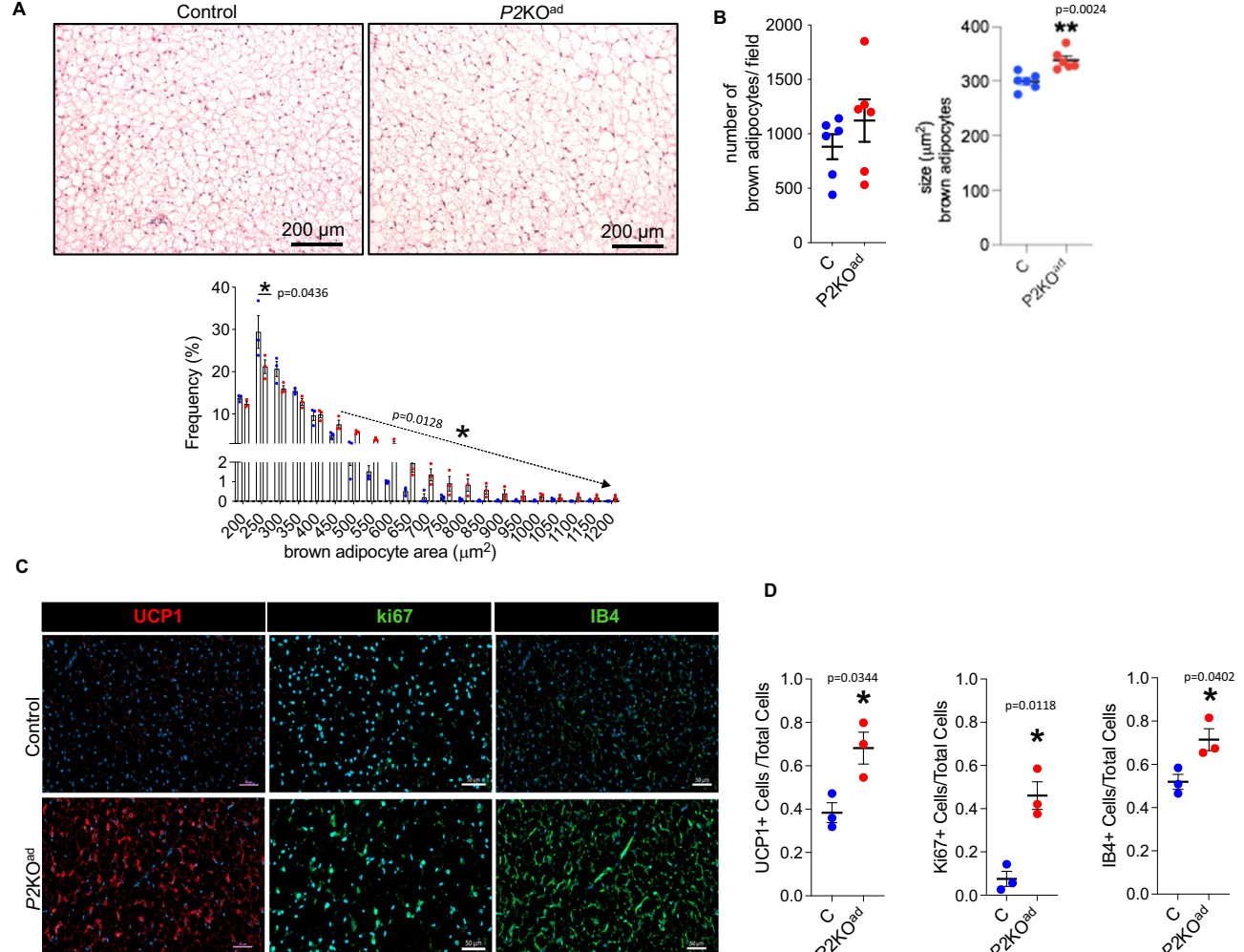

**Fig. 4 | *P2*KO^ad male mice maintain functional BAT at thermoneutrality. A** H&E staining of BAT and frequency distribution of small and large adipocytes (*n* = 3/ group, biological replicates). **B** Quantification graphs of number and size of brown adipocytes in *P2*KO^ad (red) and control littermates (blue). The adipocyte number/ size is per microscopic field (total of 6 fields per animal analysed, *n* = 6/group, biological replicates). **C** Immunofluorescence images (scale bars 50 μm) and **D** quantification graphs of BAT stained for UCP1+ (red) and ki67+ and IB4+ (green) cells at TN (*n* = 3/group, biological replicates). Nuclei stained with DAPI (blue). Data are presented as mean +/− SEM. *\*P* < 0.05, by Student *t* test (two-tailed). **A** Source data are provided as a Source Data file.

adrenergic receptor (*Adrb3)* mRNA levels was completely abolished when brown adipocytes were treated with the HIF-2a antagonist PT-2385[40], but not with the HIF1a antagonist, PX-478 (Fig. 7C), suggesting that the PHD2-HIF-2a axis is a crucial regulator of *Ucp1* expression levels. Mechanistically, HIF-2a chromatin immunoprecipitation PCR (CHiP-PCR) indicated that HIF-2a was directly associated with *Ucp1* (Fig. 7D) in brown adipocytes in hypoxia, as HIF-2a enrichment at two regions within *mUCP1* (+250, −4300) was identified.

Importantly, PHDi treatment of human abdominal subcutaneous white adipocytes from individuals with obesity increased *ADRB2* mRNA levels (Fig. 7E), a pathway recently shown to be the main target for pharmacological activation of human brown adipocytes[41]. *UCP1* mRNA expression in human adipocytes was also increased by PHDi-treatment (Fig. 7F), suggesting that FG2216 could induce a beiging-like gene signature in human WAT.

### Serum PHD2 protein levels are associated with metabolic dysfunction in humans

Our data suggest that cellular oxygen sensing proteins, and PHD2 in particular, regulate key metabolic functions and energy homeostasis when targeted in the adipocyte. We next sought genetic evidence for

the potential involvement of *PHD2* and downstream hypoxia inducible factor (HIF signalling) genes in the human metabolic phenotypes of the ~500,000 UKBioBANK genome-wide association (GWAS) database (using PhenoScanner)[42]. Genome-wide significant and/or suggestive association signals were found for blood glucose levels (*PHD2*; rs578226800, $P = 9.2E^{-08}$) and basal metabolic rate (*PHD2*; rs7534248, $P = 3.2E^{-06}$; *HIF2A*; rs11689011, $P = 9.5E^{-06}$; *HIF1AN*; rs1054399, $P = 6.9E^{-10}$). Finally, to test whether the most important known human oxygen sensor, PHD2, might serve as a target/biomarker for metabolic disease in humans, we determined if there was any association between serum PHD2 levels and metabolic syndrome traits. We analysed data acquired from a custom version of the SOMAscan proteomic profiling platform screen of serum samples from the large population-based (5457 participants) AGES-Reykjavik cohort[17,18]. The AGES Reykjavik study is a prospective study of deeply phenotyped and genotyped individuals older than 65 years of age. We found that PHD2 levels were significantly positively correlated with visceral adipose tissue (VAT via computed tomography), body mass index (BMI, kg/m²), blood markers for type 2 diabetes, HBA1C and insulin, triglycerides and the metabolic syndrome (MetS; odds ratio 1.22) (Table 1).

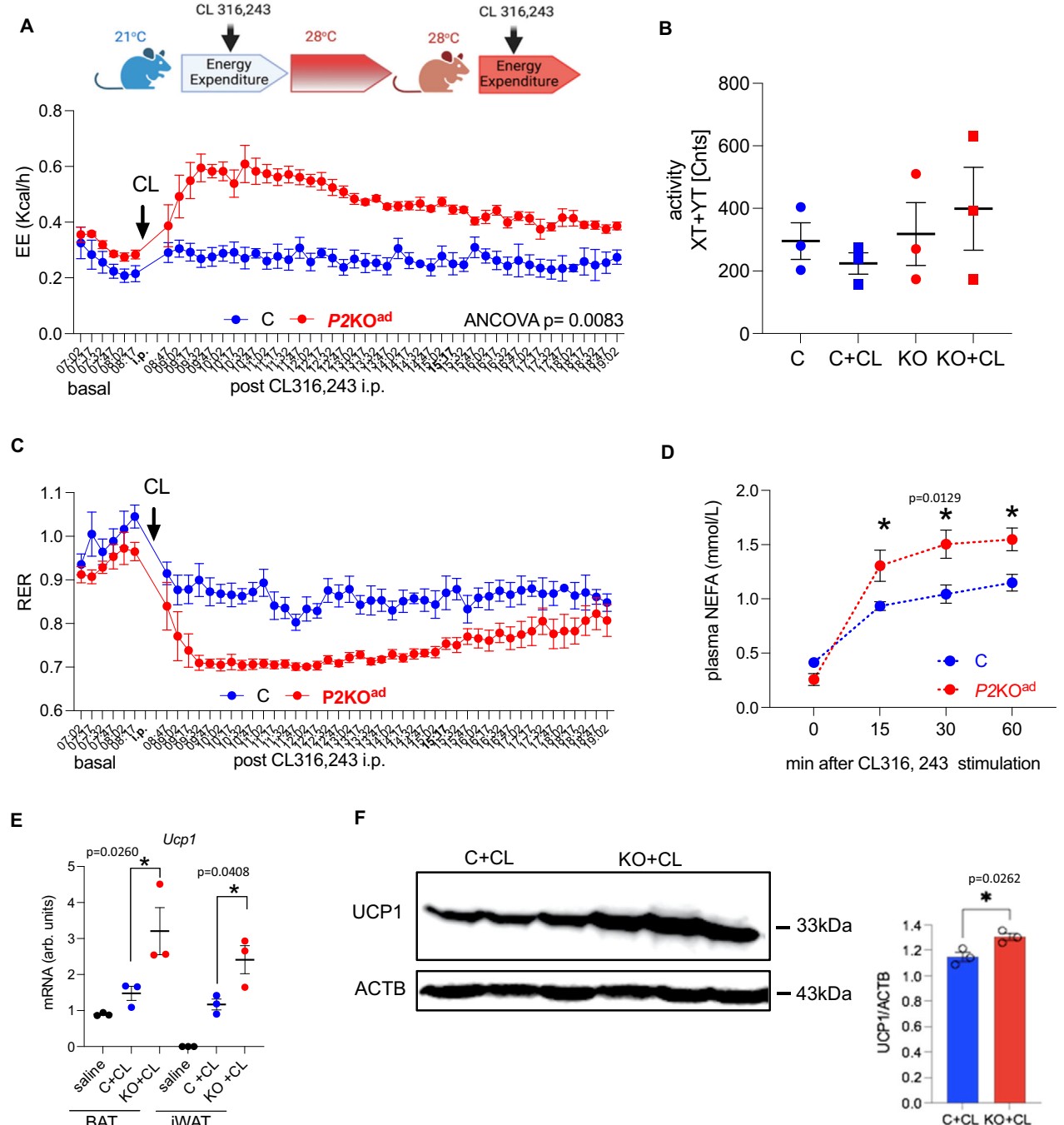

**Fig. 5 | *P2*KO^ad mice show higher sensitivity to CL316,234-induced EE at thermoneutrality.** **A** *P2*KO^ad (KO, red, *n* = 3) and control (C, blue, *n* = 3) mice were challenged with CL316,243 i.p. (CL; 1 mg/g BW) at RT (21 °C). Then switched to TN housing (28 °C) and administered second CL i.p. CL given under TN conditions led to higher energy expenditure in *P2*KO^ad mice, **B** similar activity levels in both genotypes, **C** lower respiratory exchange ratio in *P2*KO^ad mice, **D** higher sensitivity to CL-induced NEFA release in *P2*KO^ad mice, **E** higher CL-induced *Ucp1* mRNA expression in BAT and iWAT in *P2*KO^ad mice and **F** higher CL-induced UCP1 protein levels in *P2*KO^ad BAT. Data are presented as mean +/- SEM. **P* < 0.05, ***P* < 0.01 by Student *t* test (two-tailed) or one-way ANOVA (**E**). For EE, ANCOVA was performed with body weight as co-variant. A Experimental design image was created with BioRender.com released under a Creative Commons Attribution-Non Commercial-NoDerivs 4.0 International license. Source data and uncropped blots are provided as a Source Data file.

## Discussion

In this study, we have shown that the adipocyte oxygen sensing pathway regulates the thermogenic pathway by sustained β-3-adrenergic signalling and uncoupling protein (UCP)-1 expression at thermoneutrality. We postulate that this is due to enhanced responsiveness of brown adipose tissue (BAT) under conditions where BAT is normally suppressed. This is associated with increased neovascularization and maintenance of brown adipose tissue

thermogenic capacity that ultimately protects against metabolic dysfunction. The higher energy expenditure and metabolic protection was evident in both sexes and under high fat diet metabolic stress (Fig. 7G, graphical summary). We further provide evidence that human serum protein levels of PHD2 are directly correlated with measures of metabolic disease and could potentially be used as a biomarker.

Adipose tissue remodelling, in response to changing nutrient and environmental conditions, is critical to metabolic flexibility.

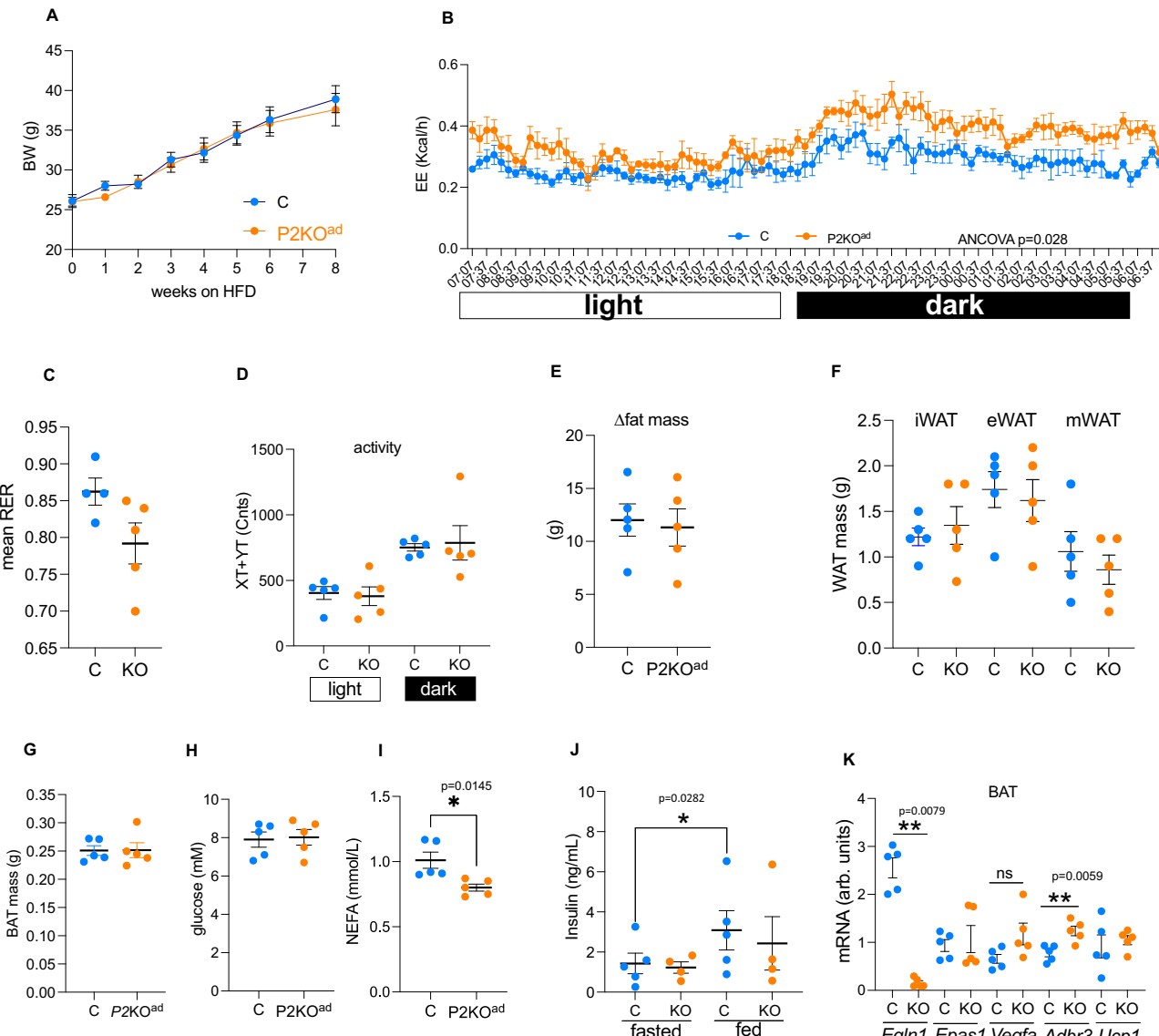

**Fig. 6 | _P2_KO[ad] mice retain higher EE after high fat feeding at thermoneutrality.** **A** _P2_KO[ad] (orange, _n_ = 5) and control (C, light blue, _n_ = 5) mice were fed high fat diet (58% kcal from fat) while housed at TN (28–29 °C) for 8 weeks had similar body weight gain. **B** Higher energy expenditure in high fat fed _P2_KO[ad] at TN. **C** Similar respiratory exchange ratio, **D** activity levels, **E** fat mass gain by TD-NMR, **F** WAT and **G** BAT weights after high fat feeding. **H** Plasma glucose, **I** NEFA and **J** insulin levels. **K** mRNA levels of HIF-target and thermogenic genes in BAT after high fat feeding at TN. Data are presented as mean + /− SEM. *_P_ < 0.05, **_P_ < 0.01, Student _t_ test (two-tailed). ns; no statistically significant. Source data are provided as a Source Data file.

Vascularization, in particular, is a crucial determinant of the oxygenation level in the tissue as recently confirmed in a human study[43]. Cifarelli et al. showed that adipose VEGFa expression was (a) significantly lower in metabolically unhealthy compared to metabolically healthy individuals with obesity and lean individuals and (b) positively associated with adipose pO$_2$ levels[43]. Our genetic model thus provides compelling evidence that deletion of _Phd2_ in adipocytes facilitates metabolic protection through multiple effects in functionally distinct adipocyte populations, including enhanced function/remodelling of BAT and enhanced lipid-retention capacity of WAT. Ultimately, this enhances whole animal energy expenditure even at thermoneutrality. However, as the mice also eat about double the amount of the controls, the outcome is that there is no difference in body weight. Thus, a caveat is-that increased metabolism does not necessarily lead to lower body weight. Metabolically, loss of adipose-PHD2 leads to lower basal lipolysis suggesting metabolic protection. It also promotes higher maximal thermogenic capacity and lipolytic capacity at thermoneutral

conditions, suggesting readiness to utilize fatty acids for thermogenesis. Although, in our pan-adipose tissue KO model, BAT activation likely drives the observed phenotype, we cannot exclude that WAT thermogenesis may also contribute to the phenotype, as we did not test a BAT (UCP1cre)-specific KO model.

In this study, we did not observe any other major phenotypic difference resulting from adipose-PHD2 loss, for example in body weight, as reported by others at RT in diet-induced obesity[22]. We postulate that this is because HFD induced the HIF-signalling response in control mice, masking subtle differences between genotypes-in effect making HIF signalling convergent under the double insult of HFD and TN. Indeed, it is well documented that HFD increases HIF1 and HIF2 levels in WAT[31,32] and we have previously shown that HFD also reduces PHD2 levels in WAT[33]. It is also evident in our current BAT data that HIF2 (_Epas1_) and the HIF-target gene (_Vegfa_) are similarly expressed in control and P2KO[ad] mice after HFD. Notably, another study, using a whole−body PHD2 inhibition, showed reduced body

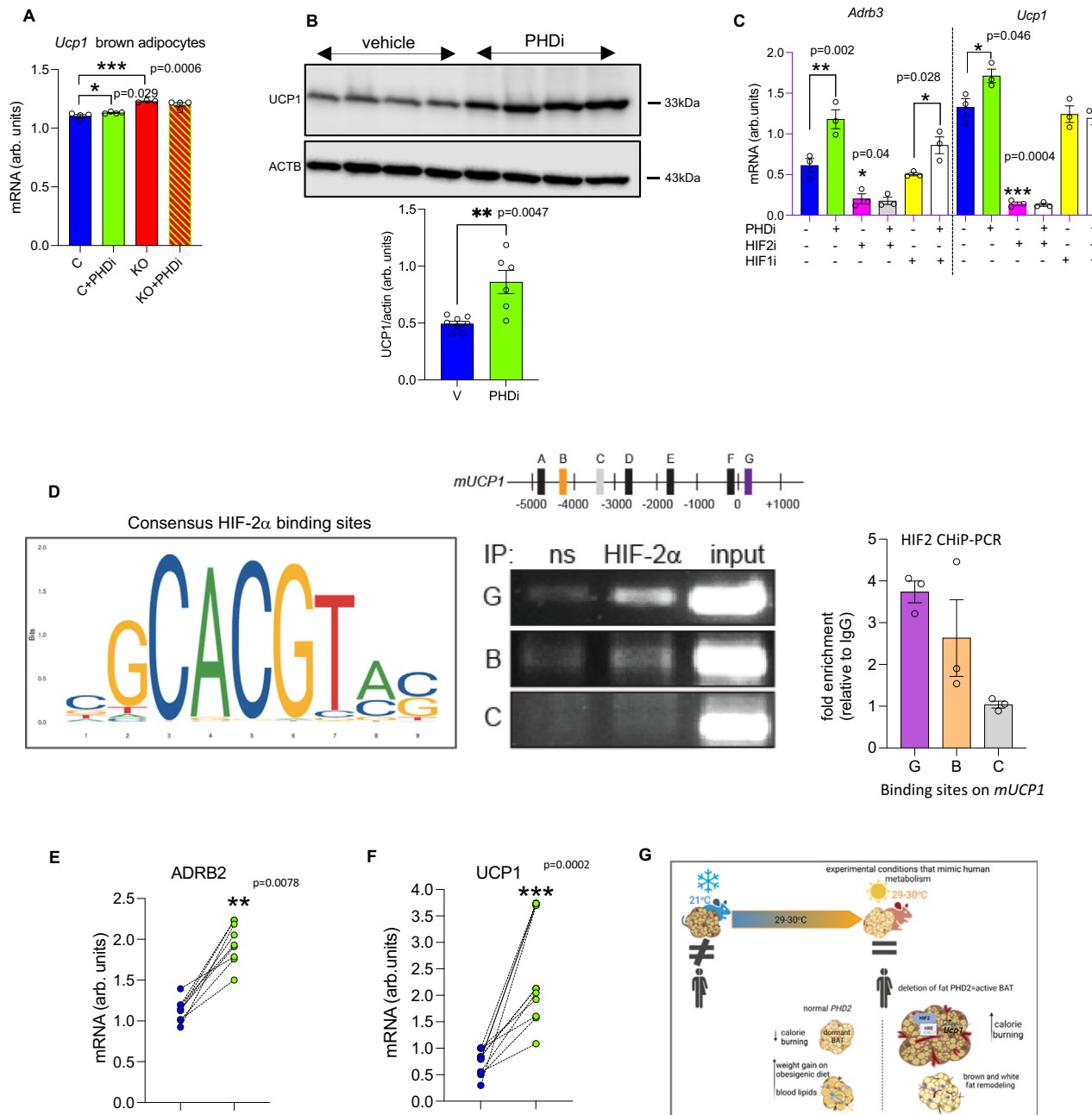

**Fig. 7 | Pharmacological pan-PHD inhibition induces *Ucp1* expression in mouse and human adipocytes in vitro. A** *P2*KO^ad (red bars) brown adipocytes have higher *Ucp1* mRNA levels than control (blue bars) mice and treatment with PHDi (FG2216, 10 μM, 16 h, *n* = 4, biological replicates) increases *Ucp1* levels only in control brown adipocytes (green bars). The mouse brown adipocyte cell line (WT-1) treated with PHDi (10 μM, 16 h, *n* = 6; green bars) shows higher **B** UCP1 protein levels (*n* = 4, biological replicates) compared to vehicle (DMSO, *n* = 4; blue bars). **C** mRNA levels of *Adrb3* and *Ucp1* are higher in the PHDi treated (green bars) WT-1 cells. Treatment with the HIF-2a antagonist (PT-2385, 10 μM, 16 h; pink bars) suppresses the effect (*n* = 3/group) in contrast to the HIF-1a antagonist (PX-478, 10 μM, 16 h; yellow bars). **D** Schematic represents *mUCP1* promoter region showing relative positions of putative HIF-2a response elements (identified using JASPAR). A–G represent putative HIF2a response elements at A; −4943, B; −4313, C; −3334, D; −2788, E; −1857, F; −120, G; +250, relative to the transcriptional start site ( + 1). ChIP performed on chromatin from WT-1 cells in hypoxia (1% O2, 6 h). PCR was performed using primer

sets that amply regions covering putative HIF-2a response elements as outlined in schematic. Lettering represents sites G; +250 (recruitment), B; −4313 (recruitment) and C; −3334 (no recruitment). ns = non-specific control immunoglobulin, IP = immunoprecipitation. Input chromatin shown (1/10 dilution relative to IP PCR). Graph represents HIF-2a fold enrichment on mUCP1 promoter compared to non-specific control immunoglobulin. **E**, **F** Human adipocytes isolated from abdominal subcutaneous biopsies treated with PHDi (10 μM, 16 h, *n* = 3 biopsies; 3 replicates per biopsy) show increased ARB2 and UCP1 mRNA expression. Data are presented as mean + /− SEM. *P < 0.05, **P < 0.01, ***P < 0.001 by Student *t* test (two-tailed). **G** illustrates that adipocyte-PHD2 deletion leads to HIF-2a activation, which in turn regulates the thermogenic pathway by sustained uncoupling protein (UCP)-1 expression and enhanced responsiveness of brown adipose tissue (BAT) under conditions where BAT is normally supressed. **G** was created with BioRender.com released under a Creative Commons Attribution-Non Commercial-NoDerivs 4.0 International license. Source data are provided as a Source Data file.

**Table 1 | Human PHD2 serum protein levels are positively correlated with metabolic syndrome related traits**

| Model[*] | Outcome variable | N | β coef. | 95% CI | S.E. | P value |
|---|---|---|---|---|---|---|
| **Linear regression** | | | | | | |
| | Body mass index, kg/m$^2$ | 5439 | 0.356 | (0.234, 0.478) | 0.062 | 1.17E-08 |
| | Visceral fat area, cm$^2$ | 5228 | 8.220 | (6.085, 10.355) | 1.089 | 5.20E-14 |
| | Triglyceride serum, mmol/L | 5447 | 0.066 | (0.048, 0.084) | 0.009 | 6.33E-13 |
| | Fasting glucose, mmol/L | 5447 | 0.063 | (0.031, 0.095) | 0.016 | 1.05E-04 |
| | HbA1c, g/dl | 5019 | 0.009 | (0.006, 0.012) | 0.001 | 4.22E-11 |
| | Insulin serum, μU/ml | 5446 | 1.127 | (0.832, 1.422) | 0.150 | 7.36E-14 |
| **Logistic regression** | | | | | | |
| | Type II diabetes | 5447 | 1.191 | (1.094, 1.297) | 0.052 | 5.76E-05 |
| | IFG (5.6–6.9 mmol/L) | 4787 | 1.057 | (0.994, 1.124) | 0.033 | 7.53E-02 |
| | Metabolic syndrome | 5443 | 1.260 | (1.186, 1.340) | 0.039 | 9.71E-14 |

*N* number of individuals with outcome data, *CI* confidence intervals, *S.E.* standard error, *IFG* impaired fasting glucose.
[*]Models were adjusted for PHD2, age and sex. PHD2 beta coefficients for continuous outcome variables are from a linear regression. For dichotomous outcome variables the PHD2 beta coefficients are odds ratios from logistic regression (see "Methods" for more details on the summary statistics). *P* values are two-sided.
Serum protein PHD2 levels were correlated with BMI, visceral adiposity, triglycerides, fasting glucose, HBA1C, insulin, type II diabetes and metabolic syndrome in the population-based Age, Gene/Environment Susceptibility (AGES) study.

weight, WAT mass and higher energy expenditure at room temperature in mice kept on a diet that specifically induced fatty liver[35]. The authors suggested that improved beigeing of WAT could explain their higher heat production[35]. In WAT, beige adipogenesis may contribute to some degree to increased thermogenesis[44,45], but it is unlikely to be the sole explanation. Pan-tissue PHD2 reduction, likely affected other tissues with a greater contribution to energy expenditure such as the skeletal muscle or brown adipose. Intriguingly, whole-body PHD2 inhibition exhibited reduced BAT mass[35] in contrast to our adipose-specific *Phd2* deletion mouse model. Effects on skeletal muscle although not studied, could potentially explain enhanced thermogenesis. For example, a recent study showed that global or skeletal muscle deletion of the PHD3 isoform led to enhanced exercise endurance capacity and a small increase in maximum oxygen consumption rate (VO2) due to increased fatty acid oxidation in muscle[46]. We cannot directly compare the phenotype of the whole-body PHD2 inhibition to our study as they were conducted under different environmental temperatures (RT vs TN) and disease models (NAFLD vs HFD). However, a consistent finding between the studies is that targeting the PHD2 isoform, and specifically in adipose tissue, may have beneficial lipid lowering effects that could be explored further in humans.

Pan-PHD1-3 inhibitors have been developed to promote the production of endogenous erythropoietin and are effective in the treatment of renal anaemia in CKD patients. The most advanced PHDi, Roxadustat, was approved for marketing in China[13] and recently also authorised for use (under the marketing name Evrenzo, Astellas Pharma Ltd/FibroGen Inc) in the European Union by the European Medicines Agency[14]. Phase 3 clinical trials in CKD patients reported that alongside the primary outcomes to increase haemoglobin and erythropoietin levels, at least some PHDi (i.e. roxadustat, dabrodustat) can lower cholesterol and low-density lipoprotein cholesterol levels[47,48]. The lowering effects on plasma fatty acids[16] and cholesterol[15,35] have also been shown in animal models treated with PHDi[49,50]. Targeting BAT activation to alleviate metabolic dysfunction, especially to lower circulating lipid levels, a hallmark of metabolically unhealthy individuals with obesity[43], is a promising new treatment line, as in both humans and mice activated BAT results in lowering of fasted and postprandial plasma triglyceride levels[51,52].

Although PHD inhibition activates both HIF1 and HIF2, HIF isoforms enact non-overlapping functions and have dominant roles in different cell types[30]. Mechanistically, our data and others[8,38,50] suggest that the PHD2-HIF2 pathway, and not HIF1, is key to brown adipose tissue regulation. Together our data supports the hypothesis that HIF2

drives transcription of *mUCP1* during hypoxia, likely via direct recruitment to HIF2 response elements within the *mUCP1* gene.

It is conceivable that PHD inhibition could be used to enhance energy expenditure. Our human adipocytes from individuals with obesity and mouse BAT cells in vitro using an earlier analogue (FG2216) closely related to roxadustat (FG4592) showed induction of β3-adrenergic signalling and UCP1 levels. These are promising findings and suggest PHDi-dependent beiging effects. However, targeting isoform specificity would be more efficacious to enhancing BAT thermogenic capacity.

Finally, our human proteomics data provide unique insight on the role of PHD2 in age-related metabolic disease. The cell/tissue origin of serum PHD2 remains to be determined, however its presence suggests that metabolic dysfunction in ageing is associated with higher PHD2 levels and possibly altered oxygen sensing. This fits with our contention that PHD2 inhibition would have beneficial metabolic effects in humans. In a timely manner, as PHD inhibitors are completing Phase 3 trials or are now approved our findings suggest that this class of drugs, if optimised for isoform and tissue targeting, could be repurposed for treatment of certain metabolic diseases.

## Limitations
We acknowledge that is not feasible to study mice strictly "at thermoneutrality". This is because the thermoneutral point changes diurnally in mice[53,54] and thus both the metabolic rate and body temperature are not constant, making it complex for assessing physiological responses in "true" mouse thermoneutral conditions. However, we chose housing at 28–29 °C to facilitate better extrapolation of our finding to human translation. The effects of increased energy expenditure and reduced blood lipids in the adipose-specific PHD2 deletion is overall consistent in both sexes housed at thermoneutral conditions, which is encouraging to consider as a target to increase energy expenditure in humans. Finally, although we identified key new evidence that HIF2 binds to the *Ucp1* promoter, providing at least one unequivocal mechanism whereby PHD2 deficiency leads to increased thermogenic gene expression, we do acknowledge that additional alternative mechanisms may also exist.

## Methods
### Study approvals
**Animal study.** All animal experiments were performed under the project licence PP5702478, appropriate PILs granted under the Home Office Scientific Procedures (Animals) Act 1983 and after full ethical review by the University of Edinburgh Biological Sciences Services.

Male and female mice were used for the experiments and were maintained single-housed in either standard or individually ventilated cages with ad libitum access to food (CRM E, Special Diets Services) and water and maintained with a 12-h light/dark cycle with lights on at 7 am, humidity 45–60%. Termination of experimental animals was done by (schedule 1 euthanasia) dislocation.

**Human study.** The Age, Gene/Environment Susceptibility Reykjavik Study (AGES-RS) was approved by the National Bioethics Committee (NBC) in Iceland (approval number VSN-00-063), which serves as the Icelandic Heart Association's institutional review board, in accordance with the Helsinki Declaration, and by the National Institute on Aging Intramural Institutional Review Board, and the Data Protection Authority in Iceland, with all participants providing written informed consent. No compensation was provided for study participation.

## Animal studies

The *Phd2* conditional allele (PHD2[f/f])[16] on a congenic C57BL/6 background (10 backcrosses), was crossed with the adiponectin–Cre allele (B6;FVB-Tg (Adipoq-cre)1Evdr/J, the Jackson Laboratories) to achieve adipose-specific conditional knockout mice (referred as *P2*KO[ad]). Genotyping PCRs were performed according to the established Jackson laboratories protocol (https://www.jax.org/Protocol?stockNumber=028020&protocolID=25488). In all experiments described, control littermates were used for comparisons. Body weight, lean and fat mass were determined by time domain–nuclear magnetic resonance (TD-NMR) (Bruker LF50; http://www.bruker.com) at RT (20–21 °C) and after 3 days at thermoneutrality (28–29 °C; TN). For the basal phenotyping control, chow diet with ad libitum access (CRM E, Special Diets Services) male and female mice (8–10 weeks old) were housed for 7 days (4 days acclimation and 3 days experimental measurements for example EE, RER, TD-NMR) in TN. Mice were single housed in indirect calorimetry cages (PhenoMaster, TSE Systems), at TN with free access to food and water, with a 12-h light and 12-h dark cycle (7 a.m. to 7 p.m.). Indirect calorimetry was used to measure energy expenditure (EE), respiratory exchange ratio (RER: $VCO_2/VO_2$), $O_2$ consumption, $CO_2$ production and physical activity (Counts/hour with X-Y-Z plane infra-red beam breaking). The EE ANCOVA analysis done for this work was provided by the NIDDK Mouse Metabolic Phenotyping Centers (MMPC, www.mmpc.org) using their Energy Expenditure Analysis page (http://www.mmpc.org/shared/regression.aspx) and supported by grants DK076169 and DK115255. For the diet induced obesity experiments, male mice (10 week old) were fed a high fat diet (58% calories as fat with sucrose; D12331, full formula can be found https://www.researchdiets.com/formulas/d12331 (Research Diets Inc, Supplementary Table S1) for 8 weeks while housed at TN. To directly test BAT function, mice were given 2× CL316, 243 (1 µg/gr BW i.p., Merck), one injection at RT and another i.p. injection after switched to TN and culled 24 h later for tissue collection. For NEFA release, mice were fasted for 4 h, a single CL316, 243 (1 µg/gr BW) i.p. was given and blood collected basally (prior to CL) 15, 30, 60 min post CL. Paired experiments were designed (as indicated in figure legends) to allow direct phenotyping comparisons of the same mouse at different temperatures. All experiments/measurements were operator/animal handler blinded with data generated by a second individual blinded to treatment until code breaking.

BAT RNA-sequencing experiment, male C57BL/6J mice were housed under controlled conditions at 22 ± 2 °C in an 12/12-h light/dark cycle maintained on a standard chow diet (Ssniff, Soest, Germany). The animals were culled for tissue collection at 10 weeks of age. Adipose tissue RNA was isolated from approximately 50 mg of ground tissue. RNA-sequencing was conducted using Illumina HiSeq 2000 with 2 ×100 bp reads (LGC Genomics, Berlin, Germany). For data processing and analysis, FASTQC v0.11.8 was used for quality assessment and trim galore v0.6.4 was applied for filtering adapter sequences and small

reads. Clean reads were mapped to a reference genome (Genome Reference Consortium Mouse Build 38 mm10) using the alignment tools HISAT2 v2.1.0 and Bowtie2 v2.3.5.1 Expression levels were quantified Cufflinks v.2.2.1.

## Blood parameters

For non-esterified fatty acids (NEFA), glycerol, glucose and insulin measurements, blood was collected after a 4 h fast. Glucose concentration was measured by a blood glucose monitoring system (OneTouchUltra2, Lifescan, Milpitas). NEFA (Wako), and glycerol (Sigma Aldrich) was measured in plasma prepared from blood collected in EDTA-coated microtubes (Sarsted). Insulin concentration was measured by a commercial Elisa kit (Crystal Chem Inc.).

## Mouse brown adipocyte culture

The WT−1 cell line was established from primary brown pre-adipocytes isolated from newborn mouse pups and immortalized with Simian virus 40 (SV40) large T antigen (38; Sigma). Briefly preadipocytes were grown in DMEM high glucose (20% FBS). A day after reaching confluence, medium was changed to induction media (Complete DMEM + IBMX 0.5 mM + Insulin 20 nM + Dexamathasone 5 µM + Indometacin 125 µM + T3 1 nM) for 2 days. Thereafter, cells were maintained in the differentiation media (Complete DMEM + Insulin 20 nM + T3 1 nM) for 5–6 days. All chemicals were from Sigma-Aldrich. Fully differentiated brown adipocytes were treated with 10 µM of 2-(1-chloro-4-hydroxyisoquinoline-3-carboxamido) acetic acid (FG-2216, Selleck Chemicals), a potent small molecule inhibitor of the PHD enzymes (PHDi) that has been shown to activate HIFα in adipose tissue[16] and/or the HIF-2a antagonist PT2385 (10 µM, MedChemExpress) and/or the HIF-1a antagonist PX-478 (10 µM, Cayman Chemical) for 16 h. Isolation of primary brown adipocytes from P2KO[ad] and control littermates was performed by collagenase digestion[16] and exposed to FG2216 (10 µM, 16 h). Adipocytes were lysed with TRIᴢᴏʟ (Invitrogen) on ice and immediately frozen for further analysis.

## RT-qPCR

Total RNA was extracted from cells and tissue using TRIᴢᴏʟ (Invitrogen) and treated with DNase I (Invitrogen). 1 µg total RNA was used for first-strand DNA synthesis using Superscript III cDNA Synthesis system (Invitrogen), and qPCR was performed with the Lightcycler 480 (Roche), using mouse-Taqman assays (Life Technologies) for all genes measured. A standard curve was constructed for each gene measured using a serial dilution of cDNA pooled from all samples. Results were normalized to the expression of *Ppia*. A list of the TaqMan gene expression assays (ThermoFisher Scientific, UK) used can be found in Supplementary Table S2.

## Chromatin immunoprecipitation (ChIP)-PCR

WT-1 cells were fully differentiated into brown adipocytes and exposed to 1% $O_2$ for 6 h prior to fixation with 1% formaldehyde for 10 min (37 °C under 1% $O_2$) followed by quenching with 0.125 M glycine. Cells were lysed in lysis I (5 mM Tris-HCl pH 8.0, 85 mM KCl, 0.5% NP-40 plus protease inhibitors) for 20 min on ice and manually disrupted using 10 strokes with Dounce homogenizer before nuclei were pelleted by centrifugation ($300 \times g$, 5 min, 4 °C). Nuclei were resuspended in nuclear lysis buffer (50 mM Tris-HCl pH 8.1, 10 mM EDTA, 1% SDS, plus protease inhibitors) on ice followed by sonication using Diagenode Bioruptor (30 cycles of 30 s on 30 s off). Samples were diluted in IP dilution buffer (0.01% SDS, 1.1% Triton x-100, 1.2 mM EDTA, 16.7 mM Tris-HCl pH 8.1, 167 mM NaCl) to achieve 0.1% SDS final, precleared for 30 min at 4 °C with blocked protein A/G (50/50 slurry of IP dilution buffer with 1 mg/mL BSA in TE, 400 µg/mL herring sperm DNA) before immunoprecipitation overnight with either HIF-2α antibody (1:250; NB100-122, Novus, Biologicals, Centennial, CO, USA) or rabbit immunoglobulin (1:250; Sigma, I5006) control. Immunoprecipitated

complexes were collected using 25 μL protein A/G slurry. Samples were washed 2× each with: wash I (20 mM Tris-HCl pH 8.1, 500 mM NaCl, 2 mM EDTA, 0.1% SDS, 1% Triton x-100), wash II (0.25 M LiCl, 1% NP-40, 1% Na-deoxycholate, 1 mM EDTA, 10 mM Tris pH 8.1), TE buffer pH 8.0. Complexes were eluted (1% SDS, 0.1 M NaHCO$_3$) for 15 min at room temperature before undergoing reverse cross-linking and RNAse digestion (65 °C for 3 h followed by overnight at 55 °C). DNA was isolated using Qiagen columns according to manufacturer's instructions. ChIP samples along with inputs (1/10 dilution) were analyzed by standard PCR using PAQ5000 hot start master mix (Agilent; Santa Clara, CA, USA, see primers in Supplementary Table S3). Quantification was performed using ImageJ/Fiji using the Gels plugin to generate lane profile plots. Results were exported to Excel and fold recruitment was calculated (IP/Ig control IP).

## Immunoblot assays

Whole tissue or cell lysates were prepared in ice-cold buffer (5 mmol/L HEPES, 137 mmol/L NaCl, 1 mmol/L MgCl$_2$, 1 mmol/L CaCl$_2$, 10 mmol/L NaF, 2 mmol/L EDTA, 10 mmol/L Na pyrophosphate, 2 mmol/L Na$_3$VO$_4$, 1% Nonidet P-40, and 10% glycerol) containing protease inhibitors (Complete Mini; Roche Diagnostics Ltd). Blots were probed with HIF-1α (1:200; 10006421; Cayman), UCP1 (1:800; PA1-24894, Thermo-Fisher) antibodies. HRP-conjugated anti-rabbit (1:1000; Dako) secondary antibody was used. Signal was detected using ECL Plus (GE Healthcare Life Sciences). Blots were re-probed with an HRP-conjugated anti-actin antibody (1:10,000; ab49900, abcam). Densitometry was performed using the ImageJ software.

## Histomorphological assessment of brown adipose

Formalin-fixed, paraffin-embedded brown adipose sections (4 μm) were used. Images were acquired using a Zeiss microscope (Welwyn Garden City, Hertfordshire, UK) equipped with a Kodak DCS330 camera (Eastman Kodak, Rochester, NY). Six randomly selected fields per BAT section in each mouse stained with H&E were captured with 10x objective. Adiposoft software (ImageJ, 1.53c) was used to determine mean adipocyte size (μm$^2$) and mean adipocyte count.

## Immunofluorescence in BAT

Paraffin-embedded BAT samples (sections 4 μm) were dewaxed by sequential incubations with xylene for 10 min, 100% ethanol for 1 min, 95% ethanol for 1 min, 80% ethanol for 1 min and 70% ethanol for 1 min. After dewaxing, slides went through an antigen retrieval step by boiling the slides in citrate buffer (0.05% Tween, pH 6) for 5 min. After fixation/dewaxing, slides were washed three times with PBS for 10 min then blocked with 10% goat serum for 1 h at RT. Sections were then incubated overnight at 4 °C with the following primary antibody: UCP1 (1:500, ab10983, Abcam), KI67 (1:50, NB500-170SS, Novus Biologicals) and isolectin B4 (1:200, B-1205, Vector Laboratories). Next day, slides were washed three times with PBS for 5 min. Subsequently, sections were incubated for 1 h at room temperature with species-specific secondary antibodies diluted 1:300. The following fluorochrome-conjugated secondary antibodies were used: anti-rabbit-Alexa 555 IgG, anti-rabbit-Alexa 647 IgG, and streptavidin conjugated 488 (all from Life Technologies). After incubation slides were washed three times with PBS for 10 min. Slides were mounted using Fluoramount G (SouthernBiotech) and images were acquired using a fluorescence microscope (Zeiss Observer, Zeiss) or a slide scanner (Zeiss, Axio Scan). Images were processed using ZEN Blue lite version (Zeiss). For quantification, random images were taken from different areas of BAT sections. The images were then split into different fluorescent channels (DAPI, AF488, AF555 or AF647) and analyzed with CellProfiler[55] to measure the median intensity of the proteins of interest (UCP1, IB4, Ki67). Data obtained from CellProfiler was then analyzed with Flowjo software or excel to obtain the number of positive cells. Negative controls images were used to gate the positive population. UCP1+ and

IB4+ are the *mean* of n = 2–3 sections/animal from three biological samples.

## Murine tissue and blood preparation for flow cytometry

Murine epididymal adipose tissue were enzymatically digested with 1 mg/ml Collagenase D (Roche) for 35 min at 37 °C in RPMI 1640 (Sigma) containing 1% foetal bovine serum (FBS) (Sigma). Blood was collected in EDTA-coated microtubes and 100 μl of blood was stained prior to red blood cell lysis with FACS lysing solution (BD). For flow-cytometry, murine cells were stained with LIVE/DEAD (Invitrogen), blocked with mouse serum and anti-murine CD16/32 (clone 2.4G2, Biolegend, 1:300) and stained for cell surface markers (Supplementary Table S4). DAPI was added to the cells prior to acquisition. All samples were acquired using a BD Fortessa LS6 and analyzed with FlowJo software (Tree Star).

## Isolation of mature human adipocytes

Ethical approval (reference number 15/ES/0094) for the collection, storage and subsequent use of human adipose tissue was granted by The Human Tissue (Scotland) Act 2006 and informed written consent was obtained from each participant. No compensation was provided for study participation. Abdominal subcutaneous adipose biopsies were collected from 3 "self-reported" females (48±4 years old) undergoing elective surgery for hernia repair or laparoscopic cholecystectomy in the Royal Infirmary of Edinburgh. Anthropometric characteristics include BMI (37.8±2.5 kg/m$^2$), waist to hip ratio (WHR: 0.88±0.023), %fat (43.3±1.8), fat mass (43.5±9 kg). Mature adipocytes were prepared from adipose tissue by collagenase digestion. Briefly, adipose tissue was immediately placed in warm KREBS buffer (Sigma-Aldrich) with 1% antibiotic/antimycotic solution, 0.5% BSA (Sigma-Aldrich) and 2 mg/ml Collagenase Type 1 (CLS-1, Worthington Biochemical) and cut into small pieces using sterile scissors and forceps and incubated for 35–45 min in shaking water bath at 37 °C. The digested samples were centrifuged at 300 × g for 5 min at 22 °C, the mature adipocytes were collected from the top portion of the centrifuge tube and transferred to a Falcon tube, washed 3× with PBS and the mature packed volume was carefully collected from the upper layer into Eppendorf tubes for immediate culture and treatment. Mature adipocytes were exposed to FG2216 (10 μM, 16 h), and immediately lysed for RNA extraction and RT-qPCR as described above.

## Human study population

Participants aged 66 through 96 are from the AGES-RS cohort[11]. AGES-RS is a single-centre prospective population-based study of deeply phenotyped subjects (5764, mean age 75 ± 6 years, 57% female "self-reported") and survivors of the 40-year-long prospective Reykjavik study (n - 18,000), an epidemiologic study aimed to understand aging in the context of gene/environment interaction by focusing on four biologic systems: vascular, neurocognitive (including sensory), musculoskeletal, and body composition/metabolism. Descriptive statistics of this cohort as well as detailed definition of the various disease relevant phenotypes measured have been published[17,18]. Definition of outcomes used in the present study: body mass index was estimated as kg/m$^2$. Visceral fat area was estimated from computed tomography images at the L4/L5 vertebrae. Area was calculated from single 10 mm trans axial images using specialized software. Blood samples were drawn after overnight fasting and centrifuged within 2 h at room temperature. Triglyceride, fasting glucose and HbA1c were analysed in serum using a Hitachi 912 analyzer (Roche Diagnostics) with reagents from Roche Diagnostics. Serum insulin was measured with a Roche Elecsys 2010 instrument. Type 2 diabetes was defined as fasting serum glucose >7.0 mmol/L or self-reported history of diabetes or the use of insulin or oral glucose-lowering drugs. Impaired fasting glucose was defined in the range 5.6–6.9 mmol/L, excluding individuals with known type II diabetes at baseline. Metabolic syndrome was defined as

meeting three of the following criteria: (i) waist/abdominal circumference >102 cm for men and >88 cm for women, (ii) triglycerides >1.69 mmol/L, (iii) high density lipoprotein <1.04 mmol/L for men and <1.30 mmol/L for women, (iv) fasting glucose >6.1 mmol/L or treated diabetes (use of antidiabetic medications−ATC group A10), (v) systolic blood pressure greater than or equal to 130 or diastolic blood pressure greater than or equal to 85 *or* treated hypertension (use of anti-hypertensive medications).

## Human serum protein measurements

Blood samples were collected at the AGES-Reykjavik baseline visit after an overnight fast, and serum prepared using a standardized protocol and stored in 0.5 mL aliquots at −80 °C. As previously described[56–58], a customized version of the SOMApanel proteomics platform was developed to include proteins known or predicted to be found in the extracellular milieu measuring 4137 unique proteins. Here for instance, PHD2 had its own detection reagent selected from chemically modified DNA libraries, referred to as Slow Off-rate Modified Aptamers (SOMAmers). The SOMAmer-based platform measures proteins with femtomole (fM) detection limits and a broad detection range or >8 logs of concentration. To avoid batch or time of processing biases, the order of sample collection and separately sample processing for protein measurements were randomized, and all samples run as a single set at SomaLogic Inc. (Boulder, CO, USA). All SOMAmers that passed quality control had median intra-assay and inter-assay coefficient of variation, CV < 5%. Hybridization controls were used to correct for systematic variability in detection and calibrator samples of three dilution sets (40%, 1% and 0.005%) were included so that the degree of fluorescence was a quantitative reflection of protein concentration. Consistent target specificity across the platform was indicated by direct (through mass spectrometry) and/or indirect validation of the SOMAmers[56]. Prior to the association analysis the protein data was centred, scaled and BoxCox transformed[59], and extreme outlier values excluded, defined as values above the 99.5th percentile of the distribution of 99th percentile cutoffs across all proteins after scaling, resulting in the removal of an average 11 samples per SOMAmer. Analyses were conducted using R version 4.2.1. and R RStudio (1.1.456).

## Statistics

All data are shown as the mean ± SEM, and a two-tailed Student's *t* test was used to compare two groups. When paired design was performed (i.e., switching housing of the same animal form RT to TN), significance was tested by paired Student's *t* test. Comparisons of 3 or more groups by one-way ANOVA. Data sets were analysed using GraphPad Prism version 8 (San Diego, California). The number of biological replicates is indicated as (n). Unpaired *t* test with Welch's correction was used for the immunofluorescence data. For the inhibitor in vitro experiments, a paired Student's test was performed to compare effects before and after drug administration in adipocytes from the same individual. A *P* value of less than 0.05 was considered statistically significant. For the associations of PHD2 to different human phenotypic measures we used linear or logistic regression depending on the outcome being continuous or binary and adjusted for age and sex in our regression analyses. Summary statistics for continuous outcome variables are listed as mean (standard deviation) and median and interquartile range (IQR) for skewed variables. For categorical variables as number and percentages *n* (%): Body mass index 27.1 (4.4), visceral fat area 172.8 (80.2), triglycerides 1.0 IQR [0.78,1.43], fasting glucose 5.8 (1.2), HbA1c 0.5 (0.1), insulin 1.2 IQR [0.79,1.78], type II diabetes 658 (12.1%), impaired fasting glucose 1982(41.4%), metabolic syndrome 1677(30.8%).

## Reporting summary

Further information on research design is available in the Nature Portfolio Reporting Summary linked to this article.

## Data availability

The data that support this study are available in the Source data file. Uncropped western blot images are available also in the Source data file. The custom-design proteomics SOMAscan is available through a collaboration agreement with the Novartis Institutes for BioMedical Research (lori.jennings@novartis.com). Data from the AGES Reykjavik study are available through collaboration (AGES_data_request@hjarta.is) under a data usage agreement with the IHA. All access to data is controlled via the use of a subject-signed informed consent authorization. The time it takes to respond to requests varies depending on their nature and circumstances of the request, but it will not exceed 14 working days. Summary statistics data for each protein's genetic determinants, i.e., protein quantitative trait loci (QTLs), have been released to a public repository (GWAS catalogue), with accession numbers detailed in Gudjonsson et al. (PMID: 35078996). Mass spectrometry data (DDA or MRM) were deposited to the ProteomeXchange Consortium with the dataset identifiers PXD008819 to PXD008823, as well as the dataset identifier PASS01145, to determine the specificity of aptamers binding to target proteins (PMID: 30072576). The mass spectrometry-based validation data for many of the aptamers included in the custom SomaScan panel used in this study are available from the PRIDE database (https://www.ebi.ac.uk/pride/) under accessions PXD008819, PXD008820, PXD008821, PXD008822, and PXD008823. The PASSEL repository (https://peptideatlas.org/passel/) under accession PASS01145. The RNA-sequencing raw data have been uploaded to NCBIs' Gene Expression Omnibus (GEO) database and can be downloaded with GEO accession number GSE269003. Source data are provided with this paper.

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

## Acknowledgements

This work was supported by the British Heart Foundation/University of Edinburgh Centre of Research Excellence Award (RE/13/3/30183 to Z.M. & RE/18/5/34216 to Z.M., T.C. and N.M.M.), Wellcome Trust ISSF3 (IS3-R3.13 21/22 to Z.M.) and a Nottingham Trent University quality related research fund (RA932 to Z.M.). C.J.S. and N.M.M. (New Investigator

award 100981/Z/13/Z) thank the Wellcome Trust for funding. I.K. is funded by the Academy of Medical Sciences (R2429101) and Rosetrees Trust (R2449101). Human adipose tissue collections were supported by a grant from the Chief Scientist Office (SCAF/17/02 to RHS) and the authors acknowledge the financial support of NHS Research Scotland (NRS), through the Edinburgh Clinical Research Facility. The authors are grateful to Professor Sir Peter Ratcliffe (Oxford University) for providing the Phd2$^{flox/flox}$ mice, valuable discussions and advice on the manuscript. The graphical summary in Fig. 7g and schematic in Fig. 5a were created with *BioRender.com*.

## Author contributions

R.W., M.G.S., I.P.C., A.C., K.F., C.B., I.K., M.M.P., and M.C. performed experiments/analysed data. R.G.M., M.B., C.J.S., R.H.S., and T.C. provided resources and/or supported with data interpretation. T.J.S. and S.G. contributed the brown adipose tissue RNA-seq data. N.M.M. designed/performed experiments. E.F.G., L.L.J., V.G., and V.E. contributed the human proteomics data. Z.M. conceived the study, acquired funding, performed experiments, supervised research and wrote the manuscript, which was reviewed by all authors.

## Competing interests

L.L.J. is an employee and stockholder of Novartis. V.E. V.G. C.J.S., N.M.M R.H.S, M.G.S, T.C., R.G.M., I.P.C, K.F., R.W., C.B., M.C., A.C., T.J.S, S.G., M.M.P., I.K., S.G., T.J.S., and Z.M. declare no competing interests.

## Additional information

[1]Centre for Cardiovascular Sciences, Queen's Medical Research Institute, University of Edinburgh, Edinburgh, UK. [2]Department of Biosciences, School of Science and Technology, Nottingham Trent University, Clifton, Nottingham, UK. [3]Department of Adipocyte Development and Nutrition, German Institute of Human Nutrition, Potsdam-Rehbrücke, Nuthetal, Germany. [4]Department of Immunology and Oncology, Centro Nacional de Biotecnología/CSIC (CNB-CSIC), Campus-UAM, Madrid, Spain. [5]Helmholtz Institute for Metabolic, Obesity and Vascular Research (HI-MAG) of the Helmholtz Zentrum München at the University of Leipzig and University Hospital Leipzig, Leipzig, Germany. [6]Chemistry Research Laboratory, Department of Chemistry and the Ineos Oxford Institute for Antimicrobial Research University of Oxford, Oxford, UK. [7]Hull York Medical School, York Biomedical Research Institute, University of York, York, UK. [8]German Center for Diabetes Research (DZD), München-Neuherberg, Germany. [9]Icelandic Heart Association, Kopavogur, Iceland. [10]Novartis Institutes for Biomedical Research, Cambridge, MA, USA. [11]Faculty of Medicine, University of Iceland, Reykjavik, Iceland. [12]Institute for Clinical Chemistry and Laboratory Medicine, Faculty of Medicine, Technische Universität Dresden, Dresden, Germany. [13]Paul Langerhans Institute Dresden, Helmholtz Zentrum München, University Hospital and Faculty of Medicine Technische Universität Dresden, Dresden, Germany. ✉e-mail: zoi.michailidou@ntu.ac.uk

