## [Peer Review File · Nature Communications]

Adipocyte deletion of the oxygen-sensor PHD2 sustains elevated energy expenditure at thermoneutralityREVIEWER COMMENTS

Reviewer #1 (Remarks to the Author):

The manuscript of Wang and colleagues, aims to demonstrate the role of the PHD2-HIF2 axis in BAT at thermoneutrality in male and female mice. Furthermore, serum PHD2 levels were correlated with increased risk of metabolic disease in humans. Although the hypothesis is of interest, as well as several of the experimental setups, the authors have failed to simply arrange the manuscript so that the text and all figures referred to match together.

Furthermore, numerous experiments only show a limited amount of individual samples (n=3) without any reference to the amount of independent repetitions. Taken together, this reviewer believes the current study is rather pre-mature with limited amount of experiments that undoubtedly underscore the different claims.

Specific issues:

- Check wrong/failing numbering of the main/supplemental figures in the text (e.g. Supplem. Fig 2E/F/G, Main Fig.3K/J is 2K/J, Supplem. 5/6, Fig5 I,J,K, L, etc...). Reviewers should not have to search for the correct figures to the written text.
- Moreover, appearance of Fig.1 I,J,K in the text comes after Fig. 2H. The reviewer suggests to group these Fig. 1 and 2 I,J,K sets as brought in the text, but maintain the color codes (Male/Fem). Alternatively, the authors could re-organize Figures 1 and 2 and discuss both males and females together in the main text. Despite certain differences the conclusion for both is similar: cKO sustains higher energy expenditure at TN.
- In the BAT of P2KOAd mice, the authors should also show the potential effect on PHD1 & 3 expression and stabilization of HIF2a. Both PHDs will be targeted with the complete PHD inhibitor at a later stage in the MS, and HIF2 is the major HIF playing a role in the studied model.
- Can the authors show reduced glycerol in males (fig 1I) as shown for females (fig2I) (RT versus TN). In supplemental Fig 1, E and F these data should refer to circulating NEFA and glycerol levels in males....
- If the authors claim that there are more brown adipocytes, then figure 3B (left panel) is not convincing (n=3 - tendency at best). Fig.3C/D UCP1 and IB4 staining in control vs cKO are not representative for the displayed quantification. Why are only 3 samples per genotype measured in Figure 3 (also in Fig 4 and 6)– how many times were these experiments repeated?
- It is unclear what data are 'not shown' in experiments under Figure 4, whilst the authors do not refer to Figure 4A in the text...
- What is the impact of weight gain over 3 days (Supplem. 5B – in the text 6B...). What conclusions do the authors want to draw from that.
- Figure 6A shows the reference protein (actin) as one line made out of 9 samples. No reliable intensity measurement per sample can be defined from this WB. The authors should demonstrate how they came to these results. How many times was this experiment repeated?
- The authors claim that the PHD2-HIF2 axis is a crucial regulator of Ucp1, using a 'specific' HIF2 inhibitor. The authors should therefore repeat the same experiments with a 'specific' HIF1 inhibitor to confirm this statement. What is the role of PHD1 and 3? Therefore, how do WT-1 cells deficient for PHD2 behave and does FG2216 still further increase the effects?

Reviewer #2 (Remarks to the Author):

Asked by the Editor to review the SOMAscan/proteomic aspects of this work, I could not find enough information to assess them. The authors state that human protein measurements were obtained from an expanded custom version of the SomaScan platform to include proteins known or predicted to be found in the extracellular milieu.

Their data sharing statement does not offer details about availability. Why are data not shared via public repositories, as typically required by Nature journals? Providing data and metadata needed to reproduce results reported in this paper, would also allow follow-up investigations, use these results of validation for future studies, and be a great resource to the community.

Details about the nature of the SOMAmers in the custom assay, QC and normalization steps, are missing altogether (except for a succinct general statement in the Supplemental Methods).

Reviewer #3 (Remarks to the Author):

In the present paper, the authors present good evidence for a remarkable metabolically enhancing effect of adipose-specific elimination of PHD2 on BAT and BAT-related parameters. These mice present with a higher amount of UCP1, resulting in a higher response to CL. They also present with a marked increase in basal metabolic rate, nearly a doubling. However, as the mice also eat about double the amount of the controls, the outcome is that there is no difference in body weight (this may be commented in the discussion in the way that it is a caveat that increased metabolism does not necessarily lead to lower body weight). There are, however, positive effects on several obesity-related parameters. The evidence for BAT recruitment is unusually convincing, and the experiments are physiologically relevantly performed (at thermoneutrality), and the metabolism is relevantly expressed per mouse (and not as divided by body weight). Generally, in these respects the paper is highly interesting.

The problem comes in understanding what is going on here, and here the authors are not clear in what is understandable or not. A normal reason for enhanced thermogenesis and BAT recruitment is constant adrenergic stimulation. However, the BAT is not chronically highly stimulated, because there is a large effect in these mice of CL, demonstrating that the tissue is far from fully activated in the absence of CL. Here may be a confounding factor in that the mice e.g. in fig 1A are stated as having been housed at thermoneutrality – but from the methods, I get the impression that the mice only have been for a few days at thermoneutrality. To state that they are “housed” at thermoneutrality demands that they have been about a month at thermoneutrality – otherwise metabolic and BAT parameters can be much influenced by the previous housing. The number of days at thermoneutrality should e.g. be stated in the figure legend. – It would also seem that several regimes have been used somewhat confusingly; e.g. in fig4 the mice are transferred during the experiment.

Concerning the data presentation, I think the authors have not been sufficiently consistent. Particularly in figs 1 and 2, I would expect directly comparable graphs – but this is not the case. One issue is again the point of dividing by body weight – an idea that I think is misleading particularly in obesity-related research (in this particular study, the issue is of less importance as such, as the body weight does not change). In fig 2E, food intake is thus

correctly given per mouse – but inconsistently, in fig 1E it is given divided by body weight. This should be altered, and all divisions by body weight altered to real values.

This about comparable graphs should also be considered when the y-axes are drawn. Here, with some exceptions (RER), I think that y-axes should start at 0, to give a correct perspective on variations etc. E.g. in fig 1A it looks like a very large variation – it is surprisingly large, particularly in the controls, but with the present representation it is exaggerated. As we are comparing the males and the females, the scales should also be the same – e.g. the fat mass/lean mass scales should be 0-25 in both cases, etc. This would also point clearer to the difference in e.g. NEFA values between the sexes. The authors may also consider harmonizing the general layout between these two figs.

The scale in fig 3B should state “number of brown adipocytes per field” (i.e. also avoiding the “total”)

For fig 4F, the tissue should be stated (I suppose it is BAT). Also, and importantly, the authors must also know the total protein amount in the tissue (a parameter that could be included in the suppl), so they can calculate the total amount of UCP1 in the tissue and include this physiologically important parameter in the fig.

What is said about the scales concerning figs 1 and 2 is of course also relevant for fig 5.

Additionally, I wonder what the scale is in fig 5K. Some kind of triple normalization yielding a mean of 1?

I am not happy with the cell line studies and the Seahorse experiments – but the authors are really not to blame for this, as they (and many with them) are following the instructions of the Seahorse company. These instructions are, however, not relevant for brown adipocyte investigations. This is simply because the cells are not stimulated, by e.g. norepinephrine or CL, and there is therefore no reason to think that the presence of any amount of UCP1 in the mitochondria of the cells would in any way affect the basal or oligomycin-inhibited respiratory rate. What perhaps is seen here is a tendency to a slightly higher mitochondrial complement in these cells, with a higher basal, coupled and leak respiration. The best would of course be if the authors redid the experiments as they should be performed, in the presence of albumin and with adrenergic stimulation, as e.g. clearly described by Li et al. (Klingenspor). A second option is to remove these results totally. A third option is to directly state – in contrast to what is presently stated in the text – that these results cannot show anything about a functional UCP1 effect, as any UCP1 would not be active under these conditions, and the leak is not UCP1-related. – I would also like to include data relating to the absolute level of UCP1 (or UCP1 mRNA) in these cells as compared to what is found in tissue in vivo.

If the authors want to point to circulating PHD2 levels as an indicator of metabolic health, it must be clear how good this parameter is. It is not sufficient to demonstrate that there – in a large population - is a statistically significant relationship between certain parameters. To ascribe interest in the relationship/correlation, the effects must also be of such relative significance that they are meaningful (contrast “the elephants in Zambia are statistically significantly 52 g heavier than those in Tanzania”). As the data presently are presented, I cannot see this.

Abstract: BAT mass is not a good indicator of BAT significance, perhaps write BAT thermogenic capacity (particularly if the authors calculate total BAT UCP1 amount as mentioned above). And the abstract may imply that it is not possible to understand what is really going on here – higher sensitivity to b3 stimulation cannot explain the high basal metabolic rate (and the authors have not demonstrated higher sensitivity, rather a higher maximal response).

Page 4: superior?

P 6: rodent-specific?

P 8: don't understand line 3.

P 8, bottom. I don't think these exp/results are meaningful. The data are "corrected" for g tissue – but the size of the cells is probably different under these conditions, and larger cells will result in lower apparent activity per g tissue, in a "correct" but meaningless way.

Probably better to remove these exps.

P10: switched to 28 -> examined at 28.

P12 top: did the authors compare to chow? – and here and elsewhere: "adipose" is not a tissue name (like liver), the expression is "adipose tissue".

P12, middle: 5L should be 5K.

P15: have the authors really shown what the claim in the first 4 lines of the discussion? And I am very doubtful to what is claimed in the second paragraph here.

P16: I don't "like" the expression "double insult" – I would definitely prefer living in the warm with good food than living in the cold with boring food. In general, I don't find the discussion section "to the point" – it seems to discuss less relevant issues than the present study itself.

Still, besides these comments, I find the results of significant interest and think that with improvements as discussed above, this could be an important contribution to the field.

Reviewer #4 (Remarks to the Author):

PHD2 deficiency in adipose tissue or the whole body has been previously reported to benefit lipid metabolism. In this manuscript, Wang et al. reported that PHD2 regulates BAT thermogenesis which is considered to be the underlying mechanism governing its regulation of lipid metabolism. Particularly, PHD2 deficient mice (P2KO) exhibit higher energy expenditure via greater BAT mass with larger brown adipocytes and higher UCP1 expression. Furthermore, the P2KO mice are more sensitive to beta3-adrenergic stimulation, indicating a greater elevation in energy expenditure (EE) and UCP1 upregulation. The reported phenotype is robust and striking. However, there remain major deficiencies in experimental design, mechanistic study and data presentation. After this study, the molecular mechanisms governing the PHD2 function on BAT activation are still not clear. The reported association between the serum PHD2 level and the increased risk of metabolic disease is of interest. However, it is not directly linked to the current study in the adipocyte-selective effect of PHD2. The deficiencies dampen the enthusiasm for this manuscript. Below are the major and minor concerns.

Major Concerns:

- 1). The current data could not sufficiently support the conclusion that activated BAT leads to increased EE in P2KO mice. They did observe the increased UCP1 and EE, which somehow support the activated BAT. However, they also observed bigger and more adipocytes in BAT, which could potentially point to impaired brown adipocyte function. Also, the morphological changes indicate the existence of other complicated regulations involving adipogenesis, lipogenesis, and lipolysis that might be involved in the overall BAT phenotype. The authors need to follow up the changes, such as the bigger lipid droplets and more cell numbers etc. and clarify how the changes regulate the metabolic status in the BAT.
- 2). Current KO model is to knock out PHD2 in both white adipose tissue and brown adipose tissue. Therefore, the metabolic phenotype, including the enhanced energy expenditure and thermogenesis is a combined effect from both fat types. The authors need to design more experiments to study the white adipose tissue (browning?) and compare the changes with in BAT.

- 3). The mechanistic study is lacking in the manuscript. Most of the results are just the characterization of metabolic phenotype. Given their previously related reports and publications from other groups, the current study only provides limited new information in the hypoxia-HIFs/PHDs research.
- 4). PHD2 KO stabilizes both HIF1alpha and HIF2alpha. These two HIFs play different roles in white adipose tissue remodeling/BAT function. The authors need to follow/discuss how both HIFs are regulated in WATs and BAT and how these different regulations affect the overall metabolic outcomes.
- 5). According to results and methods, most of the data was acquired from the mice housed at thermoneutrality (TN) unless specifically indicated shifting from RT to TN for 3 days. How long had the mice been housed at TN when mentioned "housed at TN" and what is the difference between the above-mentioned two models? What's more, while it's understandable that the authors chose to only present the data with a striking difference at TN, some other critical data that are related to BAT activation, such as the UCP1 protein level in both BAT and WAT in the RT housed as well as TN housed WT/ KO mice should also be presented. They are missing in the manuscript.
- 6). It is not clear why the authors used HIF2a antagonist to rescue the PHD inhibition effect in the cell culture model. Did the author also try HIF1a antagonist? What's the rationale for the author to choose the HIF2a antagonist over HIF1a?
- 7). The experiments including seahorse assay should be also done in the KO mice model.
- 8). What is the major organ/tissue that secretes PHD2 into circulation? How does the circulating PHD2 regulate BAT activation? The authors need to perform more experiments to address the questions.

Minor concerns.

1. The current figure organization is pretty difficult to follow. It's suggested to reorganize the figures in sequential order.
2. It is better to show the PHD2 knockout efficiency in different tissues including WAT, BAT, and other control tissues by WB.
3. Please carefully revise the panel labeling letters, especially in Sup F2.
4. Please double-check the cited literatures. Some of them are not accurate. For example, the reference for the inhibitor of the PHD enzymes in the method part is not correct.

Revised NCOMMS-22-13902A

Step-by step response to reviewers' comments

Revised manuscript with changes (text in blue)

New Figures highlighted in yellow.

Reviewer #1 (Remarks to the Author):

The manuscript of Wang and colleagues, aims to demonstrate the role of the PHD2-HIF2 axis in BAT at thermoneutrality in male and female mice. Furthermore, serum PHD2 levels were correlated with increased risk of metabolic disease in humans. Although the hypothesis is of interest, as well as several of the experimental setups, the authors have failed to simply arrange the manuscript so that the text and all figures referred to match together.

Furthermore, numerous experiments only show a limited amount of individual samples (n=3) without any reference to the amount of independent repetitions. Taken together, this reviewer believes the current study is rather pre-mature with limited amount of experiments that undoubtedly underscore the different claims.

Response: We thank the reviewer for acknowledging that our hypothesis is of interest and for comments to improve the structure of the manuscript.

Action: We apologize for any mismatch of the text and figures. We have now re-arranged the figures to ensure they match the appropriate text.

Response: power of the study (replicates): We strongly disagree with the generic statement that “*numerous experiments only show n=3/group and that the current study is rather pre-mature with limited amount of experiments that undoubtedly underscore the different claims*” for the following reasons:

(A) All the *in vivo* metabolic phenotyping work, where higher variability is expected, were done with appropriately powered group sizes (largely => n = 6/group). Importantly, in many instances, paired comparisons (before and after intervention, Fig 1. 2. 3. 5) were used which increases the statistical power. Specifically, our n numbers for the intensive *in vivo* experiments were, figure 1: n=13-15 or n=7, figure 2: n=5-9, figure 3: n=5-12, figure 6: n=5, Supplemental Fig1: n=4-7, Supplemental Fig5: n=5. 2 independent repetitions of *in vivo* experiments performed for the phenotyping at room temperature and thermoneutrality (Fig.1-2-3). *In vitro* work using cell lines was repeated in 3 independent experiments.

(B) Fig 5, CL administration, we used n=3-4/group (n=3 for control and n=4 for P2KO^{ad} RER and EE data) to test the effect of CL in a *paired* design *in vivo* under thermoneutrality. The genotype difference is clear and power calculations (<https://clincalc.com/stats/samplesize.aspx>) allowed us to draw appropriate statistical conclusions (see next page for power calculation results based for RER mean+/-SD).

Notably, there was no difference in effect of CL administration between genotypes in *room temperature* conditions. We repeated the experiment in another n=3/group to more robustly verify this (total n=6/group, see **Figure R1** below). This is possibly due to pre-existing BAT activation at 21°C. The situation is clearly different at thermoneutrality, where BAT is fully inactivated. At thermoneutrality, only the KO mice show higher CL-responses.

Specific issues:

- Check wrong/failing numbering of the main/supplemental figures in the text (e.g. Supplem. Fig 2E/F/G, Main Fig.3K/J is 2K/J, Supplem. 5/6, Fig5 I,J,K, L, etc...). Reviewers should not have to search for the correct figures to the written text.

Action: Apologies for this and thanks to the reviewer for improving our manuscript. All figure numbering has been checked stringently in the revision.

- Moreover, appearance of Fig.1 I,J,K in the text comes after Fig. 2H. The reviewer suggests to group these Fig. 1 and 2 I,J,K sets as brought in the text, but maintain the color codes (Male/Fem). Alternatively, the authors could re-organize Figures 1 and 2 and discuss both males and females together in the main text. Despite certain differences the conclusion for both is similar: **cKO sustains higher energy expenditure at TN.**

Response/Action: We thank the reviewer again for their helpful suggestions. The exact order is open to stylistic interpretation, but we have re-organised, as the reviewer suggested the original data in I, J, K (fig.1+2) into a new **Figure 3** and discusses male and female blood parameters together to present the work in a more orderly manner. Please also note that we have merged the original Fig.1 and Supplementary Fig.2 (males) into a new **Figure 1** to make the comparisons of room temperature and thermoneutrality more visually accessible. We have done a similar thing for Fig. 2 (females) and original supplementary figure 3, combining it into a new **Figure 2**. In this way, we hope, male and female data comparisons at different temperatures are clearer.

- In the BAT of P2KOAd mice, the authors should also show the potential effect on PHD1 & 3 expression and stabilization of HIF2 α . Both PHDs will be targeted with the complete PHD inhibitor at a later stage in the MS, and HIF2 is the major HIF playing a role in the studied model.

Response: We focused on PHD2, as it is widely recognized as the central oxygen sensor (PMID: 12912907; PMID: 15122348) and the primary regulator of HIF α activity and is expressed in all tissues (PMID: 18498744). The robust adipose/metabolic phenotypes of PHD2 deletion published by us and others strongly implicate PHD2 as the main isoform regulating adipocyte function.

Action: To further address the reviewer's questions, we have now included *new data* on PHD3 and PHD1 mRNA levels in BAT in P2KO^{ad} at thermoneutrality. This new data is presented in **Supplemental Fig.1A**. As expected, PHD3 is upregulated in our model, confirming findings in other models of PHD2-deletion, as PHD3 is a known hypoxia target gene (PMID: 15823097).

We have also performed a western blot for HIF2 in BAT which shows that it is stabilized in BAT in the adipose-PHD2 knockdown model. **Data in Supplemental Fig 1B.**

- Can the authors show reduced glycerol in males (fig 1I) as shown for females (fig2I) (RT versus TN).

Action: Yes, circulating glycerol is also reduced in males (was displayed in Supplemental figure 1D in the original submission), we have moved the data to **(Figure 3C)** to make it more prominent.

"In supplemental Fig 1, E and F these data should refer to circulating NEFA and glycerol levels in males.

Response: We apologise for the confusion, these data were actually NEFA and glycerol levels released into the medium of adipose explants (original version SupFig1D-E). However, under R3 request we have removed this data in the revised.

- If the authors claim that there are more brown adipocytes, then figure 3B (left panel) is not convincing (n=3 - tendency at best). Fig. 3C/D UCP1 and IB4 staining in control vs cKO are not representative for the displayed quantification. Why are only 3 samples per genotype measured in Figure 3 (also in Fig 4 and 6)– how many times were these experiments repeated?

Response & action: We have now assessed adipocyte number and size in additional samples (now total n=6/group, revised Fig.4B). There is no statistically significant difference in the number, but there is a significant difference in the size. We have adjusted the text accordingly (page 9, last 2 sentences) to refer only to differences in adipocyte size.

Fig3C/D, Apologies if it was not clear, the results on UCP1+ and IB4+ are the *mean* of n=2-3 sections/animal from 3 biological samples. We argue that the images (both in supplemental and main figures) clearly show the difference between genotypes and are representative for the quantification. We have added this information in the supplemental methods section (page 6). Importantly, the additional new data (requested by R3) of the UCP1 protein by western blot analysis in brown adipose tissue confirms higher UCP1 levels in our model (new Supplemental Fig.2D). This result is verified by 2 different methods (immunofluorescence and western blotting).

- It is unclear what data are ‘not shown’ in experiments under Figure 4, whilst the authors do not refer to Figure 4A in the text...

Response: Apologies if this was not clear. As explained above in power calculations, we have also performed the CL stimulation at room temperature as was stated in the original manuscript (page 10) “*Although, a single CL i.p. did not increase energy expenditure further at RT in either genotype (data not shown), possibly due to already active BAT at 21°C*”. We have now removed the above statement from the manuscript and the data referred to are in the rebuttal (see Figure R1 above, page 2).

- What is the impact of weight gain over 3 days (Supplem. 5B – in the text 6B...). What conclusions do the authors want to draw from that.

Response: The reviewer refers to original 5A not 5B (revised Supplemental Fig3A), weight gain is recorded after 8-weeks on HFD. The conclusion is that both genotypes gain weight equally.

- Figure 6A shows the reference protein (actin) as one line made out of 9 samples. No reliable intensity measurement per sample can be defined from this WB. The authors should demonstrate how they came to these results. How many times was this experiment repeated?

Response: In the original manuscript Fig.6 showed 9 samples (3 replicates vehicle, 6 replicates FG2216). We agree actin looks like a line, but we were able to quantify using Image J by making a consistent selection centrally over each sample, taking care not to overlap the selection between samples. There was no indication of non-equal loading (we had re-run actin twice). Below is a quantification from our excel file for the reviewer. FG2216 increased UCP1 expression. The uncropped western blot was also provided with the original

submission. A copy is provided here too, for quick reference. Note, we found a similar effect with roxadustat, another PHD inhibitor (see left hand side of uncropped WB below) and can provide this as further evidence of the UCP-1 induction effect.

line	Area -ucp1	Int density-UCP1		area actin	Int density- actin	ucp1/actin	sample id
1	1170	11728		1092	129312	0.090695	vehicle
2	1170	16560		1092	124577	0.13293	vehicle
3	1170	15494		1092	131644	0.117696	vehicle
4	1170	23251		1092	125420	0.185385	fg2216
5	1170	24711		1092	118633	0.208298	fg2216
6	1170	29714		1092	116205	0.255703	fg2216
7	1170	40351		1092	108765	0.370993	fg2216
8	1170	19005		1092	108756	0.174749	fg2216
9	1170	17248		1092	106175	0.162449	fg2216
10	1170	20635		1092	109345	0.188715	Roxadustat
11	1170	16472		1092	100083	0.164583	Roxadustat
12	1170	22593		1092	100973	0.223753	Roxadustat
13	1170	23763		1092	98386	0.241528	Roxadustat
14	1170	30647		1092	105521	0.290435	Roxadustat

Action: In the revised manuscript, we provide new western blots for both UCP1, ACTIB that confirm the original data and the actin blot is clearer. The experiment was performed 3 independent times with n=3-6/group per experiment. **New UCP1, ACTB image and quantification graph in Fig.7B.**

• The authors claim that the PHD2-HIF2 axis is a crucial regulator of Ucp1, using a 'specific' HIF2 inhibitor. The authors should therefore repeat the same experiments with a 'specific' HIF1 inhibitor to confirm this statement. What is the role of PHD1 and 3? Therefore, how do WT-1 cells deficient for PHD2 behave and does FG2216 still further increase the effects?

Response: PHD2 deletion has been shown by many studies to activate HIF1 and HF2, and the observed phenotype will be either because of HIF1 or HIF2 or both HIF1+2 activation

(PMID: 23630130; PMID: 25377876; PMID: 19217150). In terms of thermogenic response and UCP1 levels, the evidence indicates that HIF2 is important and not HIF1 as shown initially by others (and here by us) as follows:

1. Published literature:

- One of the most up-regulated genes in adipose tissue during cold exposure (1 week) is HIF2a. PMID: 19117550.
- HIF2-deletion in adipose tissue (work from co-author, Triantafyllos Chavakis) decreases UCP1 mRNA and protein levels and dampens the thermogenic response when mice are exposed to cold. (PMID: 26572826)
- The PHD2/HIF2 axis regulates catecholamines (epinephrine) synthesis and release. (PMID: 34480587) and thus is implicated in the thermogenic response.
- HIF1 deletion using the brown adipose tissue specific *Ucp1-Cre* neither affects thermogenic response, nor modulates UCP1 expression (PMID: 22302938). UCP1 expression is unaltered in adipose HIF1 transgenic models (PMID: 24906151, PMID: 24713652)

2. Importantly, our own **new** data support the above literature:

- Thus, to answer the question is “HIF1 or HIF2 responsible for the PHD2KO phenotype?” We performed pilot experiments in adipose tissue-specific (adiponectin-CRE) HIF1a KO mice, to test whether HIF1-deletion is important on the acute thermogenic response. Mice were cold exposed for 24h. Our data show that deletion of HIF1a in adipocytes does not modulate UCP1 mRNA levels, nor differentially regulates energy expenditure or circulating lipids (**Figure R2, see below, data not included in the manuscript**).
- HIF1 is not regulated by cold exposure. See below **new data in Figure R3** (this data presented in Supplemental Fig.5B), showing unaltered HIF1 levels in brown adipose tissue of C57BL6/J male mice after acute cold exposure for, 24h.
- Our data confirm that brown adipose tissue Hif2 is upregulated by acute cold, and importantly, the levels of Hif2 seen in our adipose-PHD2KO model at thermoneutrality are similar to those seen after cold exposure in control mice (**Figure R3**).
- PT2385, a highly Hif2-selective antagonist, downregulates Ucp1 mRNA levels (Fig. 7C) in the WT-1 brown adipocyte cell line and importantly the PHD inhibitor FG2216 cannot rescue this.

Action: Despite all the strong evidence above implicating HIF2 (and not HIF1), we addressed this by treating WT-1 cells with FG2216 and a HIF1 antagonist and then measure Ucp1 levels as we did in the original Fig.6 (data after HIF2 antagonism). We show in the **revised Figure 7C** that HIF1 inhibition (PX-478) does not modulate Ucp1 levels. Taken together, the lack of phenotype in HIF1KO mice after acute cold and lack of HIF1-dependent regulation of Ucp1, HIF1 is unlikely to play a role in Ucp1 regulation.

Mechanistically therefore, we suggest that it is the PHD2-HIF2 axis that drives our phenotype and not HIF1. **For all the above reasons, our decision to focus on the PHD2/HIF2 axis is justified.**

Answering the Importance of PHD2 over the other isoforms (PHD1, PHD3) in adipose tissue.

- We provide here *new* RNA-seq data on the expression levels of all the components of the HIF/PHD pathway (Phd1-2-3, Hif1-2) in BAT in mice at room temperature (**Figure R4**, this data included in **Supplemental Fig.5A**). This directly comparable data show that in the BAT of 10-week old mice housed at room temperature, Phd2 expression (red bar) is >5-fold higher than Phd1, 3 and Hif2 (light blue bar) is 10-fold higher than Hif1 expression. Also note 2 co-authors (Tim Schulz, Sabrina Gohlke, German Institute of Human Nutrition Potsdam-Rehbruecke) have been added to the revised manuscript as contributors of this data.
- Additionally in terms of PHD levels and human-adipose tissue relevance, we looked at PHD1-2-3 expression data from the Human protein atlas (<https://www.proteinatlas.org/>) and sc-RNA sequence data and it is clear that PHD2 is the most relevant and abundant isoform because:
 1. PHD2 (EGLN1) is highly expressed in metabolic tissues (skeletal muscle and adipose tissue)
 2. PHD1 (EGLN2) is mainly expressed in male tissues (testis)
 3. PHD3 (EGLN3) highest expression is seen in the heart
 4. From single cell-RNA seq data in adipocytes the expression levels of PHD2 are 10-fold higher compared to other isoforms: PHD2 (EGLN1) expression detected (65 nTPM) > PHD3 (EGLN3) expression detected (6.0 nTPM) > PHD1 (EGLN1) detected 2.8 nTPM

R1 also asks “how do WT-1 cells deficient for PHD2 behave and does FG2216 further increase the effect”.

Action: As we already had a mouse model deficient for PHD2 in brown adipocytes, we isolated primary brown fat cells from our mouse model (adiponectin cre/phd2^{flox/flox}-which we know Phd2 is >85% knocked down) and treated *in vitro* with FG2216 (to inhibit all 3 isoforms) and measured Ucp1 mRNA levels. **New Figure 7A** (a) confirms that ex-vivo brown

adipocytes that lack PHD2 have higher Ucp1 mRNA levels than control (b) inhibition of all 3 isoforms (FG2216) in control brown adipocytes increases Ucp1 and (c) Phd2 seems to be the dominant isoform, as when we inhibit all isoforms in the PHD2-KO brown adipocytes, we do not modulate further the Ucp1 response.

Taken together, all the above points and *new data provided*, indicate that the PHD2-HIF2 axis is more likely the mechanistic underpinning of the observed phenotype. We do not think that going down the route of deleting each individual PHD in the WT-1 cells is justified. There are also further experimental caveats for deletion of the isoforms in WT1 cells, such as confounding effects of deletion in the preadipocytes and not solely in mature brown adipocytes. Thus, we have focussed our efforts on providing robust evidence that PHD2-HIF2 and not HIF1 is the relevant mechanistic link.

We hope that our extensive revisions are satisfactory for the Reviewer.

Reviewer #2 (Remarks to the Author):

Asked by the Editor to review the SOMAscan/proteomic aspects of this work, I could not find enough information to assess them. The authors state that human protein measurements were obtained from an expanded custom version of the SomaScan platform to include proteins known or predicted to be found in the extracellular milieu.

Their data sharing statement does not offer details about availability. Why are data not shared via public repositories, as typically required by Nature journals? Providing data and metadata needed to reproduce results reported in this paper, would also allow follow-up investigations, use these results of validation for future studies, and be a great resource to the community.

Details about the nature of the SOMAmers in the custom assay, QC and normalization steps, are missing altogether (except for a succinct general statement in the Supplemental Methods).

Response: We have now included a more detailed description of the SOMAmer-based proteomics platform methodology and data availability to be included in the Supplemental Methods, the Data Availability section, and the Reporting Summary. We would like to point out our three recent Nature Communications publications¹⁻³, which report on many aspects of the proteomics data methodology and data availability. For example, summary statistics data was recently released to a public repository because of one of these publications¹, and we have demonstrated that protein quantitative trait loci (QTLs) identified through our proteomics platform were well replicated across different study populations and proteomic technologies^{1,3}. We note that, in contrast to the Gene Expression Omnibus (GEO) repository of mRNA expression data, there are no such databases available for the newly emerging proteomics data entering the public domain. However, for determining specificity of aptamers binding to target proteins raw mass spectrometry data (DDA or MRM) were deposited to the ProteomeXchange Consortium with the five dataset identifiers PXD008819 to PXD008823, and the dataset identifier PASS01145 as detailed in Emilsson et al.⁴ We have stated unequivocally that data sharing is done through collaboration agreement requests. Finally, individual level data cannot be released to the public domain according to Icelandic law.

The following text has now been added to Supplemental Methods:

“Serum protein measurements: Blood samples were collected at the AGES-Reykjavik baseline visit after an overnight fast, and serum prepared using a standardized protocol and stored in 0.5 mL aliquots at -80°C. As previously described¹⁻⁴, a customized version of the SOMApanel proteomics platform was developed to include proteins known or predicted to be found in the extracellular milieu measuring 4137 unique proteins. Here for instance, PDH2 had its own detection reagent selected from chemically modified DNA libraries, referred to as Slow Off-rate Modified Aptamers (SOMAmers). The SOMAmer-based platform measures proteins with femtomole (fM) detection limits and a broad detection range of >8 logs of concentration. To avoid batch or time of processing biases, the order of sample collection and separately sample processing for protein measurements were randomized, and all samples run as a single set at SomaLogic Inc. (Boulder, CO, US). All SOMAmers that passed quality control had median intra-assay and inter-assay coefficient of variation, CV < 5%. Hybridization controls were used to correct for systematic variability in detection and calibrator samples of three dilution sets (40%, 1% and 0.005%) were included so that the degree of fluorescence was a quantitative reflection of protein concentration. Consistent

target specificity across the platform was indicated by direct (through mass spectrometry) and/or indirect validation of the SOMAmers⁴. Prior to the association analysis the protein data was centred, scaled and BoxCox transformed⁵, and extreme outlier values excluded, defined as values above the 99.5th percentile of the distribution of 99th percentile cutoffs across all proteins after scaling, resulting in the removal of an average 11 samples per SOMAmer. Analyses were conducted using R version 4.2.1. and R RStudio (1.1.456)

To the Data Availability section:

“The custom-design proteomics SOMAScan is available through a collaboration agreement with the Novartis Institutes for BioMedical Research (lori.jennings@novartis.com). Data from the AGES Reykjavik study are available through collaboration (AGES_data_request@hjarta.is) under a data usage agreement with the IHA. All access to data is controlled via the use of a subject-signed informed consent authorization. The time it takes to respond to requests varies depending on their nature and circumstances of the request, but it will not exceed 14 working days. Summary statistics data for each protein's genetic determinants, i.e., protein quantitative trait loci (QTLs), have been released to a public repository (GWAS catalogue), with accession numbers detailed in Gudjonsson et al. (PMID: 35078996). Mass spectrometry data (DDA or MRM) were deposited to the ProteomeXchange Consortium with the dataset identifiers PXD008819 to PXD008823, as well as the dataset identifier PASS01145, to determine the specificity of aptamers binding to target proteins (PMID: 30072576).”

1. Gudjonsson, A., et al. A genome-wide association study of serum proteins reveals shared loci with common diseases. *Nat Commun* **13**, 480 (2022).
2. Emilsson, V., et al. A proteogenomic signature of age-related macular degeneration in blood. *Nat Commun* **13**, 3401 (2022).
3. Emilsson, V., et al. Coding and regulatory variants are associated with serum protein levels and disease. *Nat Commun* **13**, 481 (2022).
4. Emilsson, V., et al. Co-regulatory networks of human serum proteins link genetics to disease. *Science* **361**, 769-773 (2018).
5. Max Kuhn, K.J. *Applied Predictive Modeling*, (Springer, 2013).

Reviewer #3 (Remarks to the Author):

1. In the present paper, the authors present good evidence for a remarkable metabolically enhancing effect of adipose-specific elimination of PHD2 on BAT and BAT-related parameters. These mice present with a higher amount of UCP1, resulting in a higher response to CL. They also present with a marked increase in basal metabolic rate, nearly a doubling. However, as the mice also eat about double the amount of the controls, the outcome is that there is no difference in body weight (this may be commented in the discussion in the way that it is a caveat that increased metabolism does not necessarily lead to lower body weight).

Response: We thank the reviewer for this good suggestion.

Action: We have added the comment in the discussion (page 16, 2 last sentences)

2. There are, however, positive effects on several obesity-related parameters. The evidence for BAT recruitment is unusually convincing, and the experiments are physiologically relevantly performed (at thermoneutrality), and the metabolism is relevantly expressed per mouse (and not as divided by body weight). Generally, in these respects the paper is highly interesting.

Response: We thank the reviewer for their supportive comments and importantly, the acknowledgement that the temperature chosen for our studies is appropriate to study mouse thermoneutrality.

3. The problem comes in understanding what is going on here, and here the authors are not clear in what is understandable or not. A normal reason for enhanced thermogenesis and BAT recruitment is constant adrenergic stimulation. However, the BAT is not chronically highly stimulated, because there is a large effect in these mice of CL, demonstrating that the tissue is far from fully activated in the absence of CL. Here may be a confounding factor in that the mice e.g. in fig 1A are stated as having been housed at thermoneutrality – but from the methods, I get the impression that the mice only have been for a few days at thermoneutrality. To state that they are “housed” at thermoneutrality demands that they have been about a month at thermoneutrality – otherwise metabolic and BAT parameters can be much influenced by the previous housing. The number of days at thermoneutrality should e.g. be stated in the figure legend. – It would also seem that several regimes have been used somewhat confusingly; e.g. in fig4 the mice are transferred during the experiment.

Response: It is correct that for the basal phenotyping (control diet) mice were housed for 7-days (4 days acclimation and 3 days experimental measurements for example EE, RER, TD-NMR) in TN. We agree with R3, that they might not be fully adapted to TN, nevertheless, we see **(a)** expected large reduction in EE at thermoneutrality which means the protocol is working robustly and **(b)** importantly, there is a genotype effect.

Action: Therefore, we clarified this as being *an acute response* to housing at thermoneutrality (see page 7, lines 4-6). We have also added a clarification in the Supplemental Methods to be more accessible (page 1, lines 9-10). We are happy to state this as a limitation in the manuscript, if R3 wishes. **However**, we point to the key experiment with diet-induced obesity. This experimental condition was done over 8 weeks at thermoneutrality and the genotype effect on energy expenditure is clear. For this reason, we are confident that our model sustains EE in thermoneutrality (acute and more prolonged).

4. Concerning the data presentation, I think the authors have not been sufficiently

consistent.

Particularly in figs 1 and 2, I would expect directly comparable graphs – but this is not the case. One issue is again the point of dividing by body weight – an idea that I think is misleading particularly in obesity-related research (in this particular study, the issue is of less importance as such, as the body weight does not change).

Response: We have merged the blood parameters of the genders together in a new Fig3 as R1 also requested (see our response to R1, page 3).

As the reviewer points out, our model does not have a body weight phenotype, therefore, our data is not misleading. Regardless, we have updated the Figures without correction for body weight (Fig.1E, Fig.2G, Fig.2H, Fig.6F, Fig.6G).

5. In fig 2E, food intake is thus correctly given per mouse – but inconsistently, in fig 1E it is given divided by body weight. This should be altered, and all divisions by body weight altered to real values.

Response: We thank the reviewer for improving the clarity of the manuscript. We have modified for consistency.

6. This about comparable graphs should also be considered when the y-axes are drawn. Here, with some exceptions (RER), I think that y-axes should start at 0, to give a correct perspective on variations etc. E.g. in fig 1A it looks like a very large variation – it is surprisingly large, particularly in the controls, but with the present representation it is exaggerated. As we are comparing the males and the females, the scales should also be the same – e.g. the fat mass/lean mass scales should be 0-25 in both cases, etc. This would also point clearer to the difference in e.g. NEFA values between the sexes. The authors may also consider harmonizing the general layout between these two figs. The scale in fig 3B should state “number of brown adipocytes per field” (i.e. also avoiding the “total”).

Response: Thank you. All above suggestions are done in the revised.

7. For fig 4F, the tissue should be stated (I suppose it is BAT).

Response: Apologies this was not clearer, it was stated in the Figure legend.

8. Also, and importantly, the authors must also know the total protein amount in the tissue (a parameter that could be included in the suppl), so they can calculate the total amount of UCP1 in the tissue and include this physiologically important parameter in the fig.

Action: As suggested by R3, we have included this additional information as supplemental Fig.4D. All evidence from IHC, mRNA and western blot indicate higher brown adipose tissue UCP1 levels in our model.

9. What is said about the scales concerning figs 1 and 2 is of course also relevant for fig 5.

Response: Thank you, scales corrected.

10. Additionally, I wonder what the scale is in fig 5K. Some kind of triple normalization yielding a mean of 1?

Response: We are unsure what the reviewer is referring to here, in Fig.5K, the normalization was performed by calculating the ratio of gene of interest/reference gene.

11. I am not happy with the cell line studies and the Seahorse experiments – but the authors are really not to blame for this, as they (and many with them) are following the instructions of the Seahorse company. These instructions are, however, not relevant for brown adipocyte investigations. This is simply because the cells are not stimulated, by e.g. norepinephrine or CL, and there is therefore no reason to think that the presence of any amount of UCP1 in the mitochondria of the cells would in any way affect the basal or oligomycin-inhibited respiratory rate. What perhaps is seen here is a tendency to a slightly higher mitochondrial complement in these cells, with a higher basal, coupled and leak respiration. The best would of course be if the authors redid the experiments as they should be performed, in the presence of albumin and with adrenergic stimulation, as e.g. clearly described by Li et al. (Klingenspor).

A second option is to remove these results totally.

A third option is to directly state – in contrast to what is presently stated in the text – that these results cannot show anything about a functional UCP1 effect, as any UCP1 would not be active under these conditions, and the leak is not UCP1-related.

Response: We thank the reviewer, for the 3 options given, we have now removed the seahorse data, to simplify the message.

12. I would also like to include data relating to the absolute level of UCP1 (or UCP1 mRNA) in these cells as compared to what is found in tissue in vivo.

Action: As R3 requested, we have measured *Ucp1* mRNA levels in WT-1 cells, in comparison to levels found in mice in vivo under different housing temperatures. We found that fully differentiated WT-1 cells have similar *Ucp1* mRNA levels seen in brown adipose tissue of male mice housed at room temperature. This data can be found as **supplemental Figure 7** in the revised.

13. If the authors want to point to circulating PHD2 levels as an indicator of metabolic health, it must be clear how good this parameter is. It is not sufficient to demonstrate that there – in a large population - is a statistically significant relationship between certain parameters. To ascribe interest in the relationship/correlation, the effects must also be of such relative significance that they are meaningful (contrast “the elephants in Zambia are statistically significantly 52 g heavier than those in Tanzania”). As the data presently are presented, I cannot see this.

Response: This is a valid point (nice analogy). For example, an absolute measure would be beneficial, as would some sort of metric. However, could the reviewer please elaborate on what would be a satisfactory response? Currently, quantification, and thus secondary validation of serum PHD2 levels across a large population is not feasible. This represents future work.

14. Abstract: BAT mass is not a good indicator of BAT significance, perhaps write BAT thermogenic capacity (particularly if the authors calculate total BAT UCP1 amount as mentioned above). And the abstract may imply that it is not possible to understand what is really going on here – higher sensitivity to b3 stimulation cannot explain the high basal metabolic rate (and the authors have not demonstrated higher sensitivity, rather a higher maximal response).

Response: We thank the reviewer and now we refer to BAT thermogenic capacity as advised.

15. Page 4: superior?

Response: This refers to comparison of adiponectin-Cre driven to ap2-Cre driven (previously used in our Diabetes 2014 paper). We mean greater knock down of PHD2 was achieved here (with adiponectin-Cre >85% -90% whereas with ap2-Cre approximately 70%) and, unlike ap2-Cre, adiponectin-Cre is adipocyte selective (PMID: 25068087; PMID: 23321074). We removed the word superior.

P 6: rodent-specific?

Response: We mean that BAT is rodent-specific, as the amount of human BAT detected without cold stimulation is not comparable to the amount of rodent BAT.

P 8: don't understand line 3.

Response: R3 refers to “*Although when compared to male P2KO^{ad}, female P2KO^{ad} did not increase their food intake further at TN, they retained similar food intake in both temperatures (Figure 2E)*”. Apologies if it is not clear. What we mean is, that we did not see a statistically significant difference in food intake between female PHD2KO and control mice at TN. In contrast the male PHD2KO mice ate more than the controls when switched from RT to TN. We changed the wording to simplify (page 8, lines11-13).

P 8, bottom. I don't think these exp/results are meaningful. The data are “corrected” for g tissue – but the size of the cells is probably different under these conditions, and larger cells will result in lower apparent activity per g tissue, in a “correct” but meaningless way. Probably better to remove these exps.

Response: R3 refers to Supplemental data Fig.1E-F, lipolysis data in iWAT explants. Thanks, we take R3's point, given the adipocyte size difference we removed it to avoid misinterpretation.

P10: switched to 28 -> examined at 28.

Response: Corrected.

P12 top: did the authors compare to chow? – and here and elsewhere: “adipose” is not a tissue name (like liver), the expression is “adipose tissue”.

Response: Adipose tissue and circulating inflammatory cells were compared in the 2 different genotypes only after diet-induced obesity (high fat diet; HFD). We focused on the effect of HFD on immune cells as this diet triggers chronic low-grade inflammation in adipose tissue.

Corrected to “*adipose tissue*”.

P12, middle: 5L should be 5K.

Response: Apologies for mislabelling and thank you for pointing it out. Corrected accordingly.

P15: have the authors really shown what the claim in the first 4 lines of the discussion? And I am very doubtful to what is claimed in the second paragraph here.

Response: We believe the evidence we present support the claims.

P16: I don't “like” the expression “double insult” – I would definitely prefer living in the warm with good food than living in the cold with boring food.

Response: No specific comment to address.

In general, I don't find the discussion section "to the point" – it seems to discuss less relevant issues than the present study itself.

Response: We respectfully disagree with the reviewer's opinion on this. However, we did update our discussion in light of the revised data.

Still, besides these comments, I find the results of significant interest and think that with improvements as discussed above, this could be an important contribution to the field.

Response: We thank the reviewer for their overall supportive comments, and we hope that our revisions are satisfactory.

Reviewer #4 (Remarks to the Author):

PHD2 deficiency in adipose tissue or the whole body has been previously reported to benefit lipid metabolism. In this manuscript, Wang et al. reported that PHD2 regulates BAT thermogenesis which is considered to be the underlying mechanism governing its regulation of lipid metabolism. Particularly, PHD2 deficient mice (P2KO) exhibit higher energy expenditure via greater BAT mass with larger brown adipocytes and higher UCP1 expression. Furthermore, the P2KO mice are more sensitive to beta3-adrenergic stimulation, indicating a greater elevation in energy expenditure (EE) and UCP1 upregulation. The reported phenotype is robust and striking. However, there remain major deficiencies in experimental design, mechanistic study and data presentation. After this study, the molecular mechanisms governing the PHD2 function on BAT activation are still not clear. The reported association between the serum PHD2 level and the increased risk of metabolic disease is of interest. However, it is not directly linked to the current study in the adipocyte-selective effect of PHD2. The deficiencies dampen the enthusiasm for this manuscript. Below are the major and minor concerns.

Major Concerns:

1). The current data could not sufficiently support the conclusion that activated BAT leads to increased EE in P2KO mice. They did observe the increased UCP1 and EE, which somehow support the activated BAT. However, they also observed bigger and more adipocytes in BAT, which could potentially point to impaired brown adipocyte function. Also, the morphological changes indicate the existence of other complicated regulations involving adipogenesis, lipogenesis, and lipolysis that might be involved in the overall BAT phenotype. The authors need to follow up the changes, such as the bigger lipid droplets and more cell numbers etc. and clarify how the changes regulate the metabolic status in the BAT.

Response: We thank the reviewer, for the acknowledgement that “*The reported phenotype is robust and striking*”. What we have reported, in terms of brown adipose tissue histology/immunofluorescence, is that our model has bigger brown adipocytes, more brown adipose tissue vascularization, increased proliferation, and more UCP1 positive cells in BAT. Together with the whole-body in-depth metabolic phenotyping which clearly shows, in different experimental conditions (basal chow diet, under high fat diet, beta-adrenergic stimulation, both genders) higher energy expenditure in our model, supports the contention that changes in brown adipose tissue underlie the observed phenotype. We also showed the PHD2 deletion is associated with greater lipolysis by evidencing elevated BAT-activating beta-adrenergic stimulation of higher NEFA in PHD2 knockout compared to control mice. The weight of evidence supports our conclusions.

In the abstract we conclude that “Adipocyte-PHD2 deficient mice ***maintained*** higher energy expenditure at thermoneutrality via better BAT thermogenic capacity (new data in **Supp Fig 2D**, requested by R3). Our data fully support this.

The reviewer does not specify what “follow up” means beyond the extensive data already in the manuscript. We think we have made the most logical interpretation.

2). Current KO model is to knock out PHD2 in both white adipose tissue and brown adipose tissue. Therefore, the metabolic phenotype, including the enhanced energy expenditure and thermogenesis is a combined effect from both fat types. The authors need to design more

experiments to study the white adipose tissue (browning?) and compare the changes with in BAT.

Response: This study addresses the critical issue of maintained energy expenditure in thermoneutrality as a “thermally humanised” model of metabolism. Although the focus of our work is brown adipose tissue (as in mice this is a major tissue regulating the *canonical* thermogenic response), but we have provided complementary data on white adipose tissue:

- we have shown in the Supplemental Fig.1F-G (original Supplemental fig1H, fig2E), in both genders, higher expression levels of b3-adrenergic receptor and Ucp1 mRNA in subcutaneous white adipose tissue of P2KO mice.
- in Fig.5E (experiments using the b3-andrenergic receptor agonist (CL), UCP1 induction in both BAT and WAT (the latter of which suggests browning/beigeing).
- in Fig.7E,D, adipocytes from human white adipose tissue biopsies treated with a PHD inhibitor showed higher ADRB2, UCP1 expression, markers of beigeing.

We agree with the reviewer, the knock down occurs in both white and brown adipocytes, however, the phenotype of increased thermogenesis is highly likely driven by brown fat, rather than white. If we guess what the reviewer is asking for specifically, the request appears to be the wholesale design of new experiments to separate effects of BAT, WAT, thus requiring an entirely new transgenic model. While this is an interesting long term aim, it is not reasonable given the weight of evidence we provide to support our main conclusions with the current models. Besides, the dual WAT/BAT knockdown we achieve is arguably more translationally relevant to pharmacological inhibition effects that would affect all fat depots.

3). The mechanistic study is lacking in the manuscript. Most of the results are just the characterization of metabolic phenotype. Given their previously related reports and publications from other groups, the current study only provides limited new information in the hypoxia-HIFs/PHDs research.

Response: We respectfully disagree, so do the other 2 reviewers, R1 & R3 (please note R2 was only asked to comment on Soma scan data).

Some statements from the other reviewers include *“In the present paper, the authors present good evidence for a remarkable metabolically enhancing effect of adipose-specific elimination of PHD2 on BAT and BAT-related parameters. Generally, in these respects the paper is highly interesting” (R3). “The hypothesis is of interest, as well as several of the experimental setups” (R1). “I find the results of significant interest and think that with improvements as discussed above, this could be an important contribution to the field” (R3).*

In summary, our paper shows for the first time that deleting adipose PHD2 in mice causes maintained BAT mass, BAT thermogenic capacity and EE at thermoneutrality (of translational relevance to human metabolic conditions). We agree this is highly interesting and important to report. Our data show that by deleting PHD2, we activate HIF1+2, and the mechanism on thermogenesis is through the canonical, well characterised oxygen sensing/HIF pathway, but is dominantly through the HIF2 isoform-specific (rather than HIF1). Our new data also highlight the importance of the PHD2 isoform (rather than PHD1,3). Our work opens the possibility to explore translation of this pathway to increase energy expenditure in metabolic disease. PHD inhibitors (initially approved by EMA for renal anaemia) are available in UK/EU for exploring their potential use in other pathologies.

4). PHD2 KO stabilizes both HIF1alpha and HIF2alpha. These two HIFs play different roles in white adipose tissue remodeling/BAT function. The authors need to follow/discuss how both HIFs are regulated in WATs and BAT and how these different regulations affect the overall metabolic outcomes.

Response: We agree with the reviewer, that both HIF1 and HIF2 will be stabilized, and have different roles in adipose tissue remodelling. We have published extensive phenotyping on the 2 isoforms in WAT and separate roles of HIF1/HIF2 (Diabetes paper 2014). The current manuscript extends this and, as pointed out by the other reviewers, highlights interesting and important novel aspects of the role of HIF2/PHD2 in BAT and thermoneutral energy expenditure regulation. The effects in the genetic model parallel those that any potential drug targeting PHD2 would have – ie on multiple fat depots. BAT-specific models, whilst interesting, are outside the scope of the current substantial study.

Action: We have now commented in discussion (page 19, paragraph 1) regarding the evidence of HIF2 over HIF1 in brown adipose tissue regulation as this is the focus of this work.

Also, please see new **Supplemental figure 1B** (in response to R1) that shows both HIF1, HIF2 stabilization in our model and **Supplemental figures 1A** with expected target genes upregulated in BAT in our model.

5). According to results and methods, most of the data was acquired from the mice housed at thermoneutrality (TN) unless specifically indicated shifting from RT to TN for 3 days. How long had the mice been housed at TN when mentioned “housed at TN” and what is the difference between the above-mentioned two models?

Response: We thank the reviewer for asking clarification on the issue of our TN conditions. Please see our detailed response to R3 that asked for similar clarification (page 12, response to point 3).

What' more, while it's understandable that the authors chose to only present the data with a striking difference at TN, some other critical data that are related to BAT activation, such as the UCP1 protein level in both BAT and WAT in the RT housed as well as TN housed WT/ KO mice should also be presented. They are missing in the manuscript.

Response: In this manuscript we asked whether our model of adipose PHD2 deletion responds differently (especially in terms of energy expenditure) under thermoneutral housing (acute and prolonged). The data (Fig.1,2,3) clearly show that the phenotype is present under thermoneutral conditions, therefore we focused our further analysis under this condition. We have presented UCP1 positive cells by immunofluorescence (original Fig 3C, revised Fig.4C), UCP1 levels by western blot after CL-stimulation (original Fig 4F, revised Fig.5F) along with all the mRNA levels (original Supplemental Fig.1F-G; Fig.2G; Fig 3D-E/ revised supplemental Fig.1), confirming higher BAT Ucp1 protein and mRNA levels in our adipose-PHD2KO model. Please see new Supplemental Fig.1D-G for gene expression levels at room temperature and thermoneutrality.

6). It is not clear why the authors used HIF2a antagonist to rescue the PHD inhibition effect in the cell culture model. Did the author also try HIF1a antagonist? What's the rationale for the author to choose the HIF2a antagonist over HIF1a?

Response: Apologies if it was not clear, we have added the rationale in the manuscript (please see pages 13-14). We had no indication from the published literature that HIF1 is involved in the thermogenic response. Indeed, as stated above in the detailed reply to R1 (page 5-7), the evidence, including our own data, rules out HIF1 as a player in this context. Thus, pilot experiments in an adipose-specific HIF1KO model confirmed that HIF1 does not contribute to the acute thermogenic response. See response to R1 (Fig R2-R3, page 7). However, in the revised manuscript, we have also tested a HIF1a antagonist in WT-1 cells and there is no effect on UCP1 mRNA levels (revised Fig.7C).

7). The experiments including seahorse assay should be also done in the KO mice model.

Response: At the suggestion of R3, as there is ambiguity on how to interpret basal cell energetics seahorse data and UCP1 function, we will remove seahorse data from the revised manuscript to avoid misinterpretation.

8). What is the major organ/tissue that secretes PHD2 into circulation? How does the circulating PHD2 regulate BAT activation? The authors need to perform more experiments to address the questions.

Response: PHD2 is expressed in all tissues (highest levels expressed in skeletal muscle). How circulating PHD2 levels regulate BAT activation is a very interesting question but is beyond the scope of this manuscript. We do not know the answer to this, and this will require new funding and studies.

Minor concerns.

1. The current figure organization is pretty difficult to follow. It's suggested to reorganize the figures in sequential order.

Response: We thank the reviewer for encouraging us to improve our manuscript. We hope the reorganised figures (Fig1-2-3; Supplemental Fig1) have improved the clarity and presentation.

2. It is better to show the PHD2 knockout efficiency in different tissues including WAT, BAT, and other control tissues by WB.

Response: Demonstrably, the model of (adiponectin Cre-)PHD2 (Supplemental Fig.1A; >82% phd2 knockdown) works extremely well with the expected robust HIF1, 2 stabilization, and up-regulation of key hypoxia-target genes (eg Vegfa), (new additional data here in response to R1, Supplemental Fig. 1A-B) and importantly has an adipose tissue phenotype. Notably, this is consistent with the established use of the adiponectin-cre model, which drives adipocyte-specific and greater efficiency of recombination (PMID: 25068087; PMID: 23321074) than for example ap2, and so is superior to our previously characterized PHD2 deletion (using ap2cre) in our Diabetes paper 2014 in both recombination efficiency and WB data.

3. Please carefully revise the panel labeling letters, especially in Sup F2.

Response: We thank the reviewer for helping us improve our manuscript.

4. Please double-check the cited literatures. Some of them are not accurate. For example, the reference for the inhibitor of the PHD enzymes in the method part is not correct.

Response: We thank the reviewer for helping improve our manuscript. The reference in methods was chosen because we previously tested FG2216 specifically in adipose tissue and showed that HIF is stabilized. It was not intended to reference the development of the inhibitor, as it is widely commercially available. However, if the reviewer requires the original reference for the development of the inhibitor we can update.

REVIEWER COMMENTS

Reviewer #1 (Remarks to the Author):

The authors have either supported or correctly refuted the various comments and problems identified by this reviewer in the first version of this manuscript, and this in a very comprehensive manner, with interesting new experiments and data. Therefore, this revised version has convinced the reviewer to accept this manuscript for publication.

Reviewer #2 (Remarks to the Author):

The authors have addressed the comments from my previous report. I believe the manuscript is suitable for publication.

Reviewer #3 (Remarks to the Author):

The authors have in general responded adequately to my earlier comments and I do not have any important comments to the present version. I still find the basic observation of general interest but fail to understand why the extra UCP1 is activated but that will have to await further studies of this interesting model.

Reviewer #4 (Remarks to the Author):

In the revised manuscript, Wang et al. have made some improvements, addressing reviewers' concerns by incorporating new data, correcting mislabeling in figures, and refining the main text, particularly in the discussion. Despite these positive changes, several major questions persist in the paper, as outlined below:

1. In response to my previous question 1, the authors defended their findings by stating that "...in different experimental conditions..., showed higher energy expenditure in our model, supporting the contention that changes in brown adipose tissue underlie the observed phenotype." However, the model is a fat tissue (including white and brown fat tissue)-specific, not a brown fat tissue specific knockout model. Beta-adrenergic stimulation will likely induce activation in subcutaneous fat tissue too, including the upregulation of the UCP1 gene. Therefore, the conclusion that the phenotype is solely due to the activation of BAT may not be accurate.
2. Similar concern to question 1 persist in my previous question 2: energy expenditure regulation could involve both BAT and sWAT, especially under beta-adrenergic stimulation. The KO model used is fat-tissue specific, but not brown fat tissue-specific. While there is nothing inherently wrong with the animal model, the interpretations of the data are inappropriate. The data with the current model must be considered from both white fat and brown fat. To specifically address the function of BAT, a brown fat-specific KO model, such as a UCP-1 specific KO model, should be employed.
3. The authors' disagreement with my comment (previous question 3) is unclear. They cited comments from reviewers R1, R2, and R3 to defend against the question, but those comments do not address the lack of mechanistic studies in the manuscript. While the data

are “interesting and remarkable” (as reviewers commented), most of them pertain to phenotype characterization. Molecular details about how HIF2 stabilized by loss-of-PHD2 regulates thermogenic molecules, such as UCP1 are crucial for a high-impact journal submission.

4. Regarding my previous question 5, the authors asserted that "the data in fig 1, 2, 3 were generated under thermoneutral conditions, which is why we focused on this condition." However, it seems that the authors did not adequately address the question, as the rationale for focusing solely on thermoneutral conditions is unclear.

5. In response to my question 7, requesting seahorse data in the KO group, the authors stated that "we removed seahorse data per the suggestion of R3." Further clarification is needed regarding the rationale behind this decision, as it seems to have led to the omission of relevant data.

6. Concerning my question 8 about how the increased level of circulating PHD2 regulates BAT activation, the authors responded, "this will require new funding and studies," indicating a reluctance to address the question due to it being "beyond the scope of the manuscript." I disagree with this stance, as addressing this question could significantly enhance the study's significance by providing mechanistic insights into the gain-of-function of PHD2.

7. In minor question 2, I asked the authors to carefully characterize the KO efficiency of PHD2 in different fat tissues, as it might explain why they only observed a strong phenotype in BAT. Unfortunately, the authors did not address this aspect.

Overall, while the manuscript shows improvement, addressing these lingering concerns is crucial for enhancing the clarity, accuracy, and overall impact of the study.

All text revisions marked *blue* and figure changes highlighted in yellow in the **NCOMMS-22-13902B** marked copy.

NCOMMS-22-13902A

REVIEWER COMMENTS

Reviewer #1 (Remarks to the Author):

The authors have either supported or correctly refuted the various comments and problems identified by this reviewer in the first version of this manuscript, and this in a very comprehensive manner, with interesting new experiments and data. Therefore, this revised version has convinced the reviewer to accept this manuscript for publication.

Response: We thank the reviewer for acknowledging the extensive revision of our manuscript and supporting its readiness for publication.

Reviewer #2 (Remarks to the Author):

The authors have addressed the comments from my previous report. I believe the manuscript is suitable for publication.

Response: We thank the reviewer for approving our responses and supporting the manuscript for publication.

Reviewer #3 (Remarks to the Author):

The authors have in general responded adequately to my earlier comments and I do not have any important comments to the present version. I still find the basic observation of general interest but fail to understand why the extra UCP1 is activated but that will have to await further studies of this interesting model.

Response: We thank the reviewer for acknowledging that we covered their extensive requests for revision. We also note their view, in agreement with ours, that the mechanistic basis of PHD2-UCP1 would need to await further studies.

Response to R1-R2-R3:

We are delighted that that R1-R2-R3 clearly have found the revised manuscript (R1) ready for publication in *Nature Communications*, as they acknowledged our extensive revisions and our in-depth responses to their clarification requests in the rebuttal. We thank them for their time and contractive criticism to improve our manuscript.

Reviewer #4 (Remarks to the Author):

In the revised manuscript, Wang et al. have made some improvements, addressing reviewers' concerns by incorporating new data, correcting mislabeling in figures, and refining the main text, particularly in the discussion. Despite these positive changes, several major questions persist in the paper, as outlined below:

1. In response to my previous question 1, the authors defended their findings by stating that "...in different experimental conditions..., showed higher energy expenditure in our model, supporting the contention that changes in brown adipose tissue underlie the observed phenotype." However, the model is a fat tissue (including white and brown fat tissue)-specific, not a brown fat tissue specific knockout model. Beta-adrenergic stimulation will likely induce activation in subcutaneous fat tissue too, including the upregulation of the UCP1 gene. Therefore, the conclusion that the phenotype is solely due to the activation of BAT may not be accurate.

Response: We agree with the reviewer that we cannot entirely rule out a contribution from beiging/browning of white fat. To acknowledge R4s concerns, we have sentences in discussion (page 17, first paragraph, lines 5-8) stating the following:

"Although, in our pan-adipose tissue KO model, BAT activation likely drives the observed phenotype, we cannot exclude that WAT thermogenesis may also contribute to the phenotype, as we did not test a BAT (UCP1cre)-specific KO model".

We provide several lines of evidence supporting brown fat alterations as a major contributor to the phenotype, as brown fat is the dominant thermogenic tissue. This is even more plausible under thermoneutral conditions, when WAT thermogenesis would be negligible; indeed, we could not detect UCP1 in the inguinal WAT (sWAT) at thermoneutrality (please see our already deposited *Source file uncropped blots, page 3 referring to fig.5F*) where in saline group at thermoneutrality we cannot detect UCP1 protein.

Provided also here for easy access.

2. Similar concern to question 1 persist in my previous question 2: energy expenditure regulation could involve both BAT and sWAT, especially under beta-adrenergic stimulation. The KO model used is fat-tissue specific, but not brown fat tissue-specific. While there is nothing inherently wrong with the animal model, the interpretations of the data are inappropriate. The data with the current model must be

considered from both white fat and brown fat. To specifically address the function of BAT, a brown fat-specific KO model, such as a UCP-1 specific KO model, should be employed.

Response: We agree with the reviewer that although our experimental conditions support BAT thermogenesis and not WAT thermogenesis as the major contributor to thermogenesis, we cannot rule out entirely the role of WAT thermogenesis without extensive future work with additional cell-specific models. This work would however represent a new study. We have acknowledged this point *as above* (Discussion, page 17, lines 5-8), as an inherent limitation of the model (as suggested by the reviewer).

3. The authors' disagreement with my comment (previous question 3) is unclear. They cited comments from reviewers R1, R2, and R3 to defend against the question, but those comments do not address the lack of mechanistic studies in the manuscript. While the data are "interesting and remarkable" (as reviewers commented), most of them pertain to phenotype characterization. Molecular details about how HIF2 stabilized by loss-of-PHD2 regulates thermogenic molecules, such as UCP1 are crucial for a high-impact journal submission.

Response: The reviewer made the following original comment on our first manuscript: "The mechanistic study is lacking in the manuscript. Most of the results are just the characterization of metabolic phenotype. Given their previously related reports and publications from other groups, the current study only provides limited new information in the hypoxia-HIFs/PHDs research."

To address R4s original point, we responded with supporting literature and new data. The new data clarified that the predominant HIF isoform involved is HIF2A.

R4 has now requested further mechanistic insight.

Although we cannot possibly provide a full mechanistic work up in the current manuscript, we provide key evidence here that HIF2 binds to the *Ucp1* promoter, providing at least one clear mechanism whereby PHD2 deficiency leads to increased thermogenic gene expression.

Action: For this, we employed a chromatin immunoprecipitation (ChIP-PCR) assay that assessed direct HIF2 interaction with putative HREs in the *Ucp1* promoter in brown fat cells (WT-1) exposed to hypoxia. We identified seven putative HIF-2a binding sites (EPAS1 response elements) within the *mUCP1* gene (promoter and 1st exon; Figure 1, below) using JASPAR (<https://jaspar.uio.no/>) reference below for JASPAR database). To demonstrate physical binding of HIF-2a to these *mUCP1* promoter regions, we performed HIF-2a chromatin immunoprecipitation (ChIP) under hypoxia (1% O₂, 6h) followed by PCR with primer sets covering all putative sites. We identified HIF-2a enrichment at two regions within *mUCP1* (+250, -4300; Figure below). Together our data supports the hypothesis that HIF-2a drives transcription of *mUCP1* during hypoxia, likely via direct recruitment to HIF-2a response elements within the *mUCP1* gene.

This new data in the Figure below is presented in the revised Figure 7, as panel 7D.

HIF2a is recruited to the *mUCP1* promoter under hypoxia.

Schematic represents *mUCP1* promoter region (-5000 to +1000) showing relative positions of putative HIF-2a (EPAS1) response elements (identified using JASPAR).

A-G represent putative EPAS1 response elements at A; -4943, B; -4313, C; -3334, D; -2788, E; -1857, F; -120, G; +250, relative to the transcriptional start site (+1).

ChIP performed on chromatin from brown adipocytes in hypoxia (1% O₂ for 6h).

PCR was performed using primer sets that amplify regions covering putative EPAS1 response elements as outlined in schematic. Lettering represents sites G; +250 (recruitment), B; -4313 (recruitment) and C; -3334 (no recruitment).

ns = non-specific control immunoglobulin, IP = immunoprecipitation.

Input chromatin shown (1/10 dilution relative to IP PCR).

Graph represents HIF-2a fold enrichment on *mUCP1* promoter compared to non-specific control immunoglobulin.

We acknowledge in our limitations section (page 20) that alternative mechanisms may also exist as follows:

“Although we identified key new evidence that HIF2 binds to the Ucp1 promoter, providing at least one unequivocal mechanism whereby PHD2 deficiency leads to increased thermogenic gene expression, we do acknowledge that additional alternative mechanisms may also exist”.

Reference

Rauluseviciute I, Riudavets-Puig R, Blanc-Mathieu R, Castro-Mondragon JA, Ferenc K, Kumar V, Lemma RB, Lucas J, Chèneby J, Baranasic D, Khan A, Fornes O, Gundersen S, Johansen M, Hovig E, Lenhard B, Sandelin A, Wasserman WW, Parcy F, Mathelier A **JASPAR 2024: 20th anniversary of the open-access database of transcription factor binding profiles** *Nucleic Acids Res.* 2024 Jan 5;52(D1):D174–D182.; doi: [10.1093/nar/gkad1059](https://doi.org/10.1093/nar/gkad1059)

4. Regarding my previous question 5, the authors asserted that "the data in fig 1, 2, 3 were generated under thermoneutral conditions, which is why we focused on this condition." However, it seems that the authors did not adequately address the question, as the rationale for focusing solely on thermoneutral conditions is unclear.

Response: R4 originally asked "According to results and methods, most of the data was acquired from the mice housed at thermoneutrality (TN) unless specifically indicated shifting from RT to TN for 3 days. How long had the mice been housed at TN when mentioned "housed at TN" and what is the difference between the above-mentioned two models?"

We responded in detail about this, to R3s satisfaction, please see the original rebuttal (page 11 of Revision 1, response to reviewers).

Provided also here for easy access.

Response: It is correct that for the basal phenotyping (control diet) mice were housed for 7-days (4 days acclimation and 3 days experimental measurements for example EE, RER, TD-NMR) in TN. We agree with R3, that they might not be fully adapted to TN, nevertheless, we see (a) expected large reduction in EE at thermoneutrality which means the protocol is working robustly and (b) importantly, there is a genotype effect.

Action: Therefore, we clarified this as being an acute response to housing at thermoneutrality (see page 7, lines 4-6). We have also added a clarification in the Supplemental Methods to be more accessible (page 1, lines 9-10). We are happy to state this as a limitation in the manuscript, if R3 wishes. However, we point to the key experiment with diet-induced obesity. This experimental condition was done over 8 weeks at thermoneutrality and the genotype effect on energy expenditure is clear. For this reason, we are confident that our model sustains EE in thermoneutrality (acute and more prolonged).

Answering the rationale of thermoneutrality:

We thank R4 for clarifying their original question, we are sorry if we failed to understand the original meaning of it.

Our rationale is that: "This study addresses the critical issue of maintained energy expenditure in thermoneutrality as a "thermally humanised" model of metabolism (previous response, page18, lines 2-4). It is important to address energy expenditure under thermoneutrality conditions, which are the closest to the human situation.

5. In response to my question 7, requesting seahorse data in the KO group, the authors stated that "we removed seahorse data per the suggestion of R3." Further clarification is needed regarding the rationale behind this decision, as it seems to have led to the omission of relevant data.

Response: R3 provided a valid criticism relating generally to seahorse experiments in the field (PMID: 27986537). The reviewer gave us the option to remove the data so that its presentation and discussion would not lead to further confusion given the limitations their expertise reveals in the in vitro Seahorse approach to UCP activation. We realized that R3 was right and in agreement with their suggestion we removed the data.

6. Concerning my question 8 about how the increased level of circulating PHD2 regulates BAT activation, the authors responded, "this will require new funding and studies," indicating a reluctance to address the question due to it being "beyond the scope of the manuscript." I disagree with this stance, as addressing this question could significantly enhance the study's significance by providing mechanistic insights into the gain-of-function of PHD2.

Response: R4 originally asked "How does the circulating PHD2 regulate BAT activation? The authors need to perform more experiments to address the questions."

We could not have possibly guessed which experimental route to take based on the original question above. We maintain that any reasonable approach would require substantial new studies that are indeed beyond the scope of this paper. For now, we can show that circulating PHD2 in humans is, at a minimum, a potential biomarker for metabolic disease phenotypes. Establishment of causality would require in-depth and distinct preclinical studies in vitro and in vivo.

Only now does R4 suggest "gain-of-function of PHD2". Our manuscript was predominantly about the impact of PHD2 deficiency. We respectfully do not think that it is fair to raise a new comment asking for gain of function experiments (i.e. addressing the alternative hypothesis from the original scope of our article) at this stage of the review process.

7. In minor question 2, I asked the authors to carefully characterize the KO efficiency of PHD2 in different fat tissues, as it might explain why they only observed a strong phenotype in BAT. Unfortunately, the authors did not address this aspect.

Response: We did address it. We provided this information in both BAT and iWAT (Results page 6, In 11-12 and In 18-20 of the revised manuscript). PHD2 is efficiently knocked down in BAT and WAT and the expected HIF-target gene *Vegfa* is upregulated both in WAT and BAT.

That text is also pasted below for the reviewer's convenience.

"Adiponectin-Cre (23) mice were crossed with PHD2^{fl/fl} mice (16) to delete PHD2 (referred as P2KO^{ad}) in white (mean *Phd2* mRNA levels \pm SEM: Control:1.8 \pm 0.23 a.u. (n=5) vs P2KO^{ad}: 0.14 \pm 0.14 a.u. (n=6), p<0.0001) and brown adipocytes (Supplemental Figure 1A).

"*Vegfa* mRNA levels in inguinal white adipose tissue (iWAT, mean \pm SEM: Control: 0.38 \pm 0.07 a.u. (n=5) vs P2KO^{ad}: 0.73 \pm 0.12 a.u. (n=6), p=0.045) of P2KO^{ad} mice were higher compared to control littermates."

We presented the data on the 2 fat depots most relevant to this study.

Overall, while the manuscript shows improvement, addressing these lingering concerns is crucial for enhancing the clarity, accuracy, and overall impact of the study.

Response: We thank the reviewer for acknowledging our efforts. We and the other reviewers think the weight of evidence presented is of sufficient depth and quality to publish a significant advance in our field, whilst seeding important new research directions for in depth mechanism and human translational relevance.

We hope that the reviewer finds our efforts and new data satisfactory.

REVIEWERS' COMMENTS

Reviewer #4 (Remarks to the Author):

I do not have more questions.

NCOMMS-22-13902B
REVIEWERS' COMMENTS

Reviewer #4 (Remarks to the Author):

I do not have more questions.

Response: We thank the reviewer for confirming that our revisions were satisfactory.